# Mitochondrial DNA Repair in Neurodegenerative Diseases and Ageing

**DOI:** 10.3390/ijms231911391

**Published:** 2022-09-27

**Authors:** Veronica Bazzani, Mara Equisoain Redin, Joshua McHale, Lorena Perrone, Carlo Vascotto

**Affiliations:** 1Department of Medicine, University of Udine, 33100 Udine, Italy; 2Department of Advanced Medical and Surgical Sciences, University of Campania Luigi Vanvitelli, 80100 Naples, Italy; 3Department of Experimental Hematology, Institute of Hematology and Transfusion Medicine, 02-776 Warsaw, Poland

**Keywords:** mitochondria, DNA damage, DNA repair pathways, neurodegenerative diseases, Alzheimer’s disease, Parkinson’s disease

## Abstract

Mitochondria are the only organelles, along with the nucleus, that have their own DNA. Mitochondrial DNA (mtDNA) is a double-stranded circular molecule of ~16.5 kbp that can exist in multiple copies within the organelle. Both strands are translated and encode for 22 tRNAs, 2 rRNAs, and 13 proteins. mtDNA molecules are anchored to the inner mitochondrial membrane and, in association with proteins, form a structure called nucleoid, which exerts a structural and protective function. Indeed, mitochondria have evolved mechanisms necessary to protect their DNA from chemical and physical lesions such as DNA repair pathways similar to those present in the nucleus. However, there are mitochondria-specific mechanisms such as rapid mtDNA turnover, fission, fusion, and mitophagy. Nevertheless, mtDNA mutations may be abundant in somatic tissue due mainly to the proximity of the mtDNA to the oxidative phosphorylation (OXPHOS) system and, consequently, to the reactive oxygen species (ROS) formed during ATP production. In this review, we summarise the most common types of mtDNA lesions and mitochondria repair mechanisms. The second part of the review focuses on the physiological role of mtDNA damage in ageing and the effect of mtDNA mutations in neurodegenerative disorders such as Alzheimer’s and Parkinson’s disease. Considering the central role of mitochondria in maintaining cellular homeostasis, the analysis of mitochondrial function is a central point for developing personalised medicine.

## 1. Introduction

Mitochondria play a central role in the modulation of various cellular functions including energy homeostasis, proliferation, and apoptosis. Their function is crucial for ATP production, synthesis of heme and steroid hormones, calcium and iron homeostasis and subsequent signalling cascades [1], fatty acid oxidation, and the regulation of interorganelle contacts with the endoplasmic reticulum (ER) [2]. The mitochondrial network is characterised by a dynamic structure aimed at rapidly responding to the energy demands of the cell [3]. Mitochondria possess their own DNA and are characterised by specific mechanisms regulating the mitochondrial network: the turnover of mitochondrial DNA (mtDNA) [4], the highly regulated mitochondrial fission and fusion, as well as the mitophagy that eliminates dysfunctional mitochondria [5]. The pathways regulating mitochondrial turnover and homeostasis are called mitochondrial quality control (MQC) and consist of DNA repair mechanisms, reactive oxygen species (ROS) scavenging, chaperones and proteolytic enzymes, the ubiquitin–proteasome system (UPS), the mitochondria-specific unfolded protein response (UPRmt), mitochondrial fusion and fission dynamics, and mitochondrial biogenesis and degradation (Figure 1) [6]. Mitochondrial injury impairs cellular function and exerts a multiorgan effect. Neurons are particularly susceptible to mitochondrial dysfunction because of their high energetic demand that mostly depends on oxidative phosphorylation (OXPHOS). Notably, the brain makes up for about 2% of the body mass but expends about 20% of the oxygen and energy of the whole body [7]. In addition, neurons are characterised by a unique architecture including extremely long processes (axons can be longer than 1 m in humans) and by being extremely long-lived postmitotic cells that cannot efficiently dilute out defective organelles by cell division, making these cells very sensitive to energy alterations [8]. Recent studies have revealed that age-dependent impaired energy metabolism in neurons precedes neurodegeneration [8]. In agreement, mitochondrial dysfunction is an early event in neurodegenerative diseases as it promotes brain decline caused by ageing [9]. Neurodegenerative diseases are characterised by progressive atrophy and loss of neuronal function. Although several familiar forms of neurodegenerative diseases have been characterised by analysing the function of mutant genes, the relevance of altered metabolic function as a key factor promoting the onset and progression of sporadic neurodegenerative diseases is emerging [10]. In this context, mitochondrial alterations promote key events occurring during ageing and in neurodegeneration. In this review, we focus on the crucial role of mtDNA maintenance and alterations in promoting neurodegeneration. Since mitochondria, together with the nucleus, are the only organelle possessing their own DNA, it is relevant to point out the role of environmental factors that may affect the integrity of mtDNA and, in turn, participate in neurodegeneration. In agreement with this, mitochondria have developed several mechanisms necessary to maintain the integrity of their own DNA [11]. On the other hand, alterations in mtDNA cause metabolic dysfunction that, in turn, affects the neurons [12]. Herein, we describe the effect of mtDNA homeostasis failure in promoting ageing and neurodegenerative processes.

## 2. Oxidative Stress and Mitochondrial DNA Lesions

### 2.1. Point Mutations and Ribonucleotide Incorporation

It is has been extensively demonstrated that mtDNA mutates with higher frequency compared to nuclear DNA (nDNA) because of mtDNA’s enhanced exposure to ROS such as O_2_^−^ or H_2_O_2_, which can be produced during OXPHOS system [17], the lack of histones, and a less efficient repair system for mtDNA damage compared to the nucleus [18]. 

Oxidative stress represents an imbalance between the production of ROS and their elimination by protective mechanisms. In physiological conditions, ROS can activate diverse signalling cascades [19] and are involved in cellular processes such as proliferation [20], apoptosis [21], and senescence [22]. Antioxidant enzymes and ROS scavenger systems such as superoxide dismutases (SODs), the thioredoxin system, and glutathione peroxidase counteract ROS production, thus protecting the cells from the dangerous effects associated with an imbalance in ROS production [23]. When there is an imbalance between ROS production and the antioxidant systems, the resulting oxidative stress can damage all the main cellular components. Indeed, oxidative stress activates various pathways including the proinflammatory cascade and promotes the formation of promutagenic DNA adducts creating genetic instability that leads to DNA mutations, which alter cellular homeostasis [24]. Consequently, oxidative DNA lesions and lipid peroxidation-derived DNA lesions are common in mitochondria with 8-oxo-7,8-dihydro-2′-deoxyguanosine (8-oxo-dG) being the most widely used marker for mtDNA damage. 

mtDNA is constituted by an H-strand and an L-strand, which are both translated and encoded for mitochondrial proteins. Spontaneous point mutations can arise on both mtDNA strands. These strands are differentially affected by spontaneous deamination giving rise to G→A/T→C transition mainly in the L-strand and C→T/A→G transition in the H-strand. The frequency of this spontaneous phenomenon increases with age [25].

Finally, during replication, ribonucleotides (rNMPs) can be incorporated not only in nDNA but also in mtDNA, where they are better tolerated [26]. Although the misincorporation of rNMPs can be ascribed to erroneous replication by the DNA polymerase γ (POLγ), the major cause of the high frequency of rNMPs in mtDNA is the lack of proficient rNMP repair systems in mitochondria. Indeed, the isoform of RNase H1 present in mitochondria is not sufficient to compensate for the lack of RNase H2, which is the enzyme responsible for removing single embedded rNMPs in the nucleus [27].

### 2.2. Deletions

The presence of mtDNA deletions has been reported since the late 1980s. Deletions in mtDNA derive from alterations in two main mechanisms: (i) mtDNA replication; (ii) mtDNA repair impairment resulting in double-strand breaks. It has been also hypothesised that these two mechanisms cooperate in producing mtDNA deletions. The majority of the mtDNA deletions occur in the major arc of the mtDNA and have been associated with different pathologies, where the clinical prognosis directly correlates with mtDNA deletion frequency. Nonetheless, specific deletions are also found on the minor arc [28]. The most prevalent deletion is called common deletion (CD) and is associated not only with pathologies—it was first described in a patient with Kearns-Sayre syndrome [29]—but also with ageing [30]. Alongside the majority of mtDNA lesions, CD levels are linked to mosaicism [31], which influences the phenotype depending on the anatomical location and the tissue affected by the deletion. 

### 2.3. Single-Strand and Double-Strand DNA Breaks

Single-strand breaks (SSBs) and double-strand breaks (DSBs) are discontinuities in one or two strands of the DNA, respectively, that can occur on mtDNA directly (e.g., from attack of ROS) or indirectly (e.g., during the enzymatic cleavage of the phosphodiester backbone mediated by BER). The presence of these lesions is lower compared to point mutations and deletions, likely because they are eliminated by a specific clearance mechanism.

## 3. Mitochondrial DNA Repair Pathways and Coping Mechanisms

### 3.1. The Base Excision Repair Pathway

The first and best characterised mechanism of DNA repair described in mitochondria is the BER pathway [32]. Oxidative damage caused by ROS is repaired by BER, which is characterised by 3 steps: (1) recognition and excision of the damaged DNA base; (2) removal of the resulting abasic (AP) site; and (3) gap filling and ligation (Table 1). 

Step (1) is performed by a DNA glycosylase, which catalyses the cleavage of the *N*-glycosidic bond between the damaged base and its deoxyribose. In step (2), the resulting AP site can be recognised by an AP endonuclease (APE1), which hydrolyses the phosphate backbone (the same function can be performed by the glycosylase itself). There are two categories of glycosylases: monofunctional and bifunctional. Monofunctional glycosylases lack the lyase activity and rely on APE1. Bifunctional glycosylases possess a lyase activity and are able to create a 3′ nick after the removal of the damaged base. Moreover, monofunctional and bifunctional glycosylases recognise different DNA alterations: alkylation and deamination are the main target of monofunctional glycosylases while oxidised bases are repaired by bifunctional glycosylase activity [33]. In the final step (3), POLγ is recruited to incorporate the correct nucleotide and a ligase completes the process, ligating the previously formed nick. When BER is activated, it can follow two subpathways: the short-patch or the long-patch BER (SP- or LP-BER) [34]. The main difference is related to the number of nucleotides that are substituted during the correction process and the proteins involved. In the SP-BER, only the damaged nucleotide is removed and corrected, while in the LP-BER, from two up to eight nucleotides surrounding the damaged base can be substituted during the repair process.

The SP-BER steps are common in nuclei and mitochondria, but they differ slightly in the enzymes involved. Particularly, in mitochondria, only one DNA ligase has been described, DNA ligase III, which is involved in both DNA replication and repair. DNA ligase I is also present in nuclei but not in mitochondria [35]. More interesting is the study of the LP-BER subpathway in mitochondria. Until recently, it was believed that only the SP-BER was active within mitochondria. However, several studies carried out in the last decade clearly indicated the existence of a mitochondrial LP-BER, wherein the protein FEN-1 plays a crucial role [36,37,38]. 

### 3.2. The Mismatch Repair Pathway and Double-Strand Break Repair Pathways

The presence of the mismatch repair (MMR) pathway in mitochondria is still unclear. Among the proteins involved in this pathway, only YB-1 has been identified in mitochondria, suggesting the presence of MMR involving the protein of the BER system. Indeed, recent studies demonstrated that YB-1 interacts with the glycosylase NEIL2, APE1, and the DNA ligase III (Table 1) [39]. However, more research is necessary in order to confirm this pathway. 

mtDNA is subjected to DSBs just as the nDNA, but in mitochondria, the mechanisms involved in DSBs repair are not yet fully elucidated. Evidence of mtDSB repair has been found in Drosophila [40] and Saccharomyces cerevisiae [41]. However, the presence of proteins such as XRCC1 [42] or an alternate form of Ku80 [43] in mitochondria is not sufficient to confirm the capacity of mitochondria to repair their DSB through homologous recombination (HR) or nonhomologous end joining (NHEJ).

Some studies suggest that recombination occurs more frequently intermolecularly than intramolecularly [44]. Interestingly, other reports support the role of a not-well-characterised microhomology-mediated end-joining (MMEJ) repair pathway in mtDNA repair [45,46,47] rather than the NHEJ, which appears to be undetectable in mitochondria. This hypothesis is held by the detection of short repetitive sequences flanking the deletions in mtDNA occurring in 85% of Drosophila older than 55 days and in two-thirds of the reported mitochondrial deletions of ageing humans [48,49]. This suggests the existence of a recombination mechanism involved in the maintenance of mtDNA integrity.

### 3.3. mtDNA Degradation

A single mitochondrion may contain multiple copies of mtDNA. Some mtDNA molecules can carry lesions and give rise to a heterogeneous pool of mtDNA. This phenomenon is called heteroplasmy. A heteroplasmic cell undergoing mtDNA replication—which is independent from the cell cycle—can accumulate mutations over time [50]. When a heteroplasmic cell undergoes cell cycle, it gives rise to a mosaic distribution of the lesion. When the level of lesions overcome a certain threshold, the lesion manifests itself as a mitochondrial dysfunction. Typically, this threshold is higher than 80%, implying that most mtDNA mutations are haploinsufficient or recessive [51].

The selective depletion of mtDNA is a phenomenon driving the control of the amount of mutated mtDNA. In a single cell, there are thousands of mtDNA molecules present and the clearance of only some of these molecules does not compromise mitochondrial function [11]. It is, therefore, plausible that lesions such as DSBs can be eliminated via mtDNA degradation [52]. The mechanisms underlying mtDNA degradation are not yet fully elucidated. Two scenarios have been proposed: (i) degradation by nucleases [53] and (ii) elimination of the whole mitochondria carrying the lesion via mitophagy [54,55]. mtDNA degradation is activated in the presence of excess DNA damage. This mechanism is not damage-specific and its kinetics vary depending on the cell type [56]. Recently, it has been shown that the linear DNA formed upon DSBs appears to be degraded via the exonuclease activity of POLγ and MGME1 [57]. Nevertheless, the role of other mtDNA replication enzymes such as Twinkle helicase [58] and mtSSB [59] in this pathway is still debated (Table 1).

In 2020, Xiuli Dan and colleagues demonstrated the involvement of mitophagy in the quality control of mitochondrial integrity [60]. They showed that mitophagy is induced following DNA damage and its activation is independent of the stressor triggering the damage. They also emphasised the critical role of Spata18 in this process, shedding light on the mechanism of mitophagy initiation. At early timepoints following DNA damage, it appears that cells will preserve, or attempt to preserve, enough mitochondria in order to support the cell’s energy demands. However, this is in contrast to later timepoints, where the rate of mitophagy is increased, and cells will clear any damaged or unnecessary mitochondria. 

Overall, it has been postulated that following irreparable mtDNA damage, mitochondrial fission is stimulated, leading to mitophagy of the damaged mitochondrial daughter.

### 3.4. Mitochondrial Dynamics

Mitochondria form a dynamic network of organelles able to fuse and divide in two phenomena known as fusion and fission, respectively. This dynamic is supported by several proteins, which coordinate this process in a very accurate manner. Mitochondrial fission is mediated primarily by the GTPase dynamin-related protein 1 (Drp1), whose mitochondrial recruitment is controlled by numerous mitochondrial outer membrane receptors such as Fis1, Mff, MiD49, and MiD51 [61]. Mitochondrial fusion is realised through mitofusin 1 (Mfn1), Mfn2, and Opa1, three GTPases of the dynamin superfamily [62]. Mitochondrial fission and fusion are critical for the ability of the cell to cope with damaged mtDNA. Bulky adducts that arise on mtDNA cannot be repaired by the nucleotide excision repair (NER) pathway such as within the nucleus. Indeed, there is no clear evidence of the existence of this pathway in mitochondria [63]. Indeed, mtDNA lesions such as pyrimidine dimers, base modifications or inter-/intrastrand and DNA-protein cross-links are cleared by isolating dysfunctional mitochondria and their removal by selective mitochondrial fusion and mitophagy [64]. However, the molecular mechanism is still unclear. Moreover, whether mtDNA damage and mitochondrial fission dynamics are correlative or consequential is also elusive. Indeed for both fusion and fission, it is not clear if the processes start because of stress and, so, independently of the mtDNA damage or if it is a consequence of the damage detected on the mtDNA. On the other hand, decreased Drp1 activity and its delocalisation contributes to neurodegeneration by promoting mitochondrial dysfunction [65]. More studies are required to elucidate this phenomenon and its involvement in the quality control of mtDNA. 

**Table 1 ijms-23-11391-t001:** Mechanisms involved in mtDNA repair. The best characterised DNA repair pathway in mitochondria is the BER pathway. Several publications have investigated the role of the BER machinery in mtDNA maintenance and the difference to its nuclear counterpart. Attempts to elucidate the presence of other mtDNA repair mechanisms have been performed with contradictory results. Here, we enlist proteins known to be involved in different nuclear repair pathways and that have been studied for their role in mtDNA integrity and stability. Underlined text represents enzymes with exclusive mitochondrial localisation. DSBs = double-strand DNA breaks; HR = homologous recombination; LP-BER = long-patch base excision repair; MMR = mismatch repair; NHEJ = nonhomologous end joining; SSBs = single-strand DNA breaks.

Repair Pathway	Lesion	Enzyme	Enzyme Class	Function	Note	Ref.
BER	Oxidative damage	MUTHY AGG UNG	Hydrolase	Monofunctional glycosylase		[66]
		OGG1 NTH		Bifunctional glycosylase (β-elimination)		
		NEIL1 NEIL2		Bifunctional glycosylase (βδ-elimination)		
		APE1 PNK	Hydrolase	Hydrolysis of phosphate backbone		[67,68]
		Polγ	Transferase	Nucleotide incorporation		[69]
		FEN-1	Hydrolase	Cleavage of 5′ flap structures	LP-BER	[70]
		ExoG		Removal of 5′-blocking moiety	LP-BER	[71]
		DNA ligase III	Ligase	Nick ligation		
MMR	Base mismatches	YB-1	DNA-binding protein	Mismatch sensing and protein recruitment		[39,72]
HR/NHEJ	SSBs and DSBs	No evidence of HR/NHEJ activity in mammalian mitochondria				
mtDNA degradation	Any lesion	MGME1	Hydrolase	5’–3’ exonuclease activity	Involved in degradation of linear DNA after DSBs	[73]
		Twinkle	Helicase	Unwinding of mtDNA replication fork	Potentially involved in mtDNA degradation	[74,75]
		mtSSB	Single-strand DNA binding protein	Enhancing Twinkle and Polγ activity		[11,59]

## 4. The Role of mtDNA in Ageing

Ageing is a physiological process whose basic mechanisms are generally conserved from yeast to humans and can be classified into three categories: primary, antagonistic, and integrative [76]. Primary features include the main causes associated with ageing damage such as mitochondrial dysfunction due to altered mtDNA [77]. Antagonistic and integrative features, on the other hand, focus on the response to the damage and its consequences (Figure 2) [77]. The focal point of this section is to give an overview of the role of mitochondria and the increasing interest in the role of this organelle in the ageing processes. Indeed, different theories have been proposed over the years and are summarised here [78,79,80]. All these approaches show both limitations and strengths and a comprehensive theory explaining the mitochondrial role in ageing is still missing. Nevertheless, the study of this organelle in the context of ageing is crucial considering that ageing is a risk factor for a plethora of diseases and can severely affect the prognosis of many patients.

Multiple factors are involved in ageing and can be classified in three categories: (1) primary factors that cause the occurrence of damage; (2) antagonistic factors related to the response of the cell to the damage; and (3) integrative factors, culprits of the phenotype. All these phenomena are connected and, over time, lead to ageing. In this scenario, mitochondria also have an important role both as primary (mtDNA damage) and/or antagonistic factors (mitochondrial dysfunction).

### 4.1. Mitochondrial Free Radical Theory of Ageing

In 1956, Dr. Denham Harman proposed the mitochondrial free radical theory of ageing (MFRTA) [81]. Harman’s theory starts from the simple observation that free radicals may cause oxidative damage through attacks on cell interior mechanisms inducing the degeneration of cells and tissues within the body. This damage could occur to nucleic acids and genetic material, leading to mutations causing cancers, and may also be a contributing factor to ageing. Harman concluded that with key chemical control, life itself could then be prolonged. In following publications, Harman suggested that these free radicals were produced within mitochondria due to their high oxygen usage and the correlation between basal metabolic rate and ageing. He went on to allude that ageing originates within mitochondria as well, with mtDNA receiving around 16 times more oxidative damage than nDNA [82]. 

In 2005, de Grey suggested that in nondividing cells such as neurons, oxidative damage occurs at similar rate as in other cells but with less degradation of mitochondria. Thus, these dysfunctional mitochondria are able to proliferate and accumulate over the lifetime of nondividing cells leading to cellular dysregulation, impaired respiratory capacity, and promotion of the ageing phenotype [83]. Direct evidence for this theory is still lacking. On the contrary, various studies demonstrate that somatic mutations in mtDNA occur independently of oxidative stress [84,85]. Indeed, mutations that impair SOD2 activity have no effect on mtDNA mutation frequency in vivo [84], suggesting that the in vitro studies supporting de Grey’s theory are not sufficient to unveil the mechanisms occurring in vivo. In addition, other studies show that increased oxidative stress, induced by glucose restriction or by hypoxia, is beneficial due to activation of cellular resistance to stress resulting in reduced mortality in cell and laboratory animal models, and in diabetic patients [86,87]. All together, these data suggest that mtDNA mutations and alterations promote ageing in a manner independent of oxidative stress. 

### 4.2. Clonal Expansion Theory

A link between mtDNA deleterious mutations and ageing was first suggested in 1988 [88]. Related studies showed that low levels of common deletions in mtDNA exist in tissues, particularly in skeletal muscle. These deletions accumulate within mtDNA through clonal expansion [89]. Clonal expansion is a well-characterised process in B-lymphocytes whereby a large number of cells can be selected for yielding a particular genotype characterised by specific DNA deletions. This mechanism has been translated into a theory of mitochondrial dysregulation leading to ageing called “clonal expansion theory” [90]. Positive selection pressures can occur within cells, particularly for cancer-promoting nDNA mutations [91,92]. Interestingly, mtDNA mutations have been associated with age through increased numbers of mutated mitochondria building up rather than individual mitochondria becoming increasingly damaged. The presence of random mutations even in young healthy individuals supports this hypothesis [93]. Specific point mutations and deletions in mtDNA are present during early development and even in germline cells [90,94]. In agreement, mtDNA mutations are present in those tissues known to appear very early during embryonic development [95]. These mutations lead to a deficiency in mitochondrial OXPHOS, inducing metabolic alterations within proliferating cells and, thus, decreasing apoptosis. This decreased apoptosis results in dysfunctional and deficient mitochondria remaining prevalent within cells as well as promoting premature ageing. In fact, mitochondrial OXPHOS deficiency is accepted as a hallmark of ageing [96]. In addition to promoting ageing, the mutations are implicated in the promotion of cancer in proliferating cells while also inducing senescence in nonproliferating cells [97].

### 4.3. The Gradual ROS Response Theory

The gradual ROS response theory postulates that ageing itself is a loss of balance within homeostasis caused by the accumulation of unspecific ROS-dependent damage [80]. Under normal conditions and through organismal development, the stress defences and antioxidants respond to ROS production and counteract its effects. However, over a lifetime, the human body develops ROS-independent damage, which stimulates the cell stress response, thereby elevating ROS production in a vicious cycle. Indeed, when this elevation in ROS overwhelms the antioxidant system, ROS directly damage the pathways responsible for the maintenance of cellular homeostasis. This process finally produces the ‘ageing phenotype’. This theory explains the correlation between elevated ROS levels and senescence. Such oxidative damage can cause additional stress responses, which further enhance ROS production, leading to a vicious cycle of cell damage that accelerates ageing and mortality. From this viewpoint, it may be possible to hypothesise that ageing is a beneficial evolutionary process in most organisms due to selection pressures and adaptation to the damage mechanisms described above [98]. 

### 4.4. Ageing and Mitochondria: One Theory to Rule Them All

Considering the theories developed in the last decades, it has become clear that ROS are implicated in ageing and mitochondria contribute to this process through mechanisms that are not yet fully characterised. Up to now, it has been difficult to propose a comprehensive picture of the molecular mechanisms responsible for the ageing process. An interconnection of pathways and biological mechanisms promote the slow, but inexorable, decline of the human organism. Although all three theories described above propose relevant concepts, the validity of each of them is still debated. Indeed, some reports support the beneficial effect of the ROS scavenger therapy in promoting longevity [99], while other studies present the opposite results [100]. Considering the complexity of ageing, we can hypothesise that mtDNA damage represents one of a number of mechanisms involved in ageing and can be more relevant in some organisms compared to others. For the greater understanding of the complexity of ageing, it is critically important to continue and extend the investigation of mtDNA damage. This investigation is also crucial for the unveiling of mechanisms that participate in the progression of age-related diseases. Below, we focused on the role of mtDNA alterations in neurodegenerative diseases.

## 5. The Role of mtDNA in Neurodegenerative Diseases

Neurodegenerative diseases (NDs) can be either hereditary or sporadic conditions, which define a broad range of heterogeneous disorders, caused by the progressive degeneration of specific neurons in the central (CNS) and peripheral nervous system (PNS) [101]. A common feature of NDs is the initial mild condition such as, but not exclusively, coordination or memory problems due to initial neuronal dysfunction, followed over time by more serious impairments that compromise everyday life as a consequence of neuronal loss [102]. Unfortunately, NDs are fatal, and their prevalence has been increasing in recent years [103]. Herein, we focus on the most recent studies concerning the NDs affecting the CNS.

Alzheimer’s disease, Parkinson’s disease, amyotrophic lateral sclerosis (ALS), and Huntington’s disease are the most known and better characterised NDs. However, the molecular mechanisms implicated in these pathologies are not yet fully elucidated. In the last two decades, studies have revealed the pathological involvement of oxidative stress and its crucial effect on the integrity of DNA [104,105,106]. It is still unclear whether oxidative stress plays a causative role in NDs or if it is a consequence due to the degenerative process. However, several lines of evidence suggest that it has an impact on both the onset and the progression of NDs. Indeed, several studies focus on the role of mitochondrial dysfunctions and oxidative damages on mtDNA integrity in the context of ND [107]. Herein, we summarised the up-to-date knowledge about the role of mitochondria and mtDNA damage in ND progression (Figure 3); AD and PD haplogroups are summarised in Table 2.

### 5.1. Alzheimer’s Disease

Alzheimer’s disease (AD) is the most common cause of dementia among older people and is characterised by the occurrence of extracellular amyloid deposits (senile plaques) of the amyloid beta (Aβ) peptide and by intraneuronal aggregates of hyperphosphorylated tau protein in structures named neurofibrillary tangles (NFTs) [108]. While about 5–10% of AD cases are familiar due to mutations in genes related to Aβ production, the majority of AD cases are sporadic and develop from complex interactions among genes, environment, and epigenetic and stochastic factors [109]. The impact of each of these factors on AD progression is still unclear, and it is possible that there is a synergy among these aspects in promoting AD [110]. Several studies investigated the impact of mtDNA damage in AD, but the results were contradictory [111]. However, there is evidence confirming that the presence of mtDNA damage in AD leads to energy failure, increased oxidative stress, and Aβ formation, which, in turn, exacerbates mtDNA damage and oxidative stress promoting a vicious circle of damage. Nonetheless, up to now, there is no evidence demonstrating a causative role of mtDNA mutations in AD [112]. On the other hand, various data underline the role of BER components acting in mitochondria. Post mortem brains of AD patients present decreased levels of 5-hydroxyuracil (5OHU) incision as well as diminished DNA ligase III activity, suggesting an impaired function of the DNA repair pathway, which can have negative consequences on the overall quality of mtDNA [113].

Finally, an interesting association has been revealed between mtDNA polymorphisms and AD. Small differences in the encoded proteins can slightly affect the OXPHOS activity leading to either overproduction or reduction of free radicals. Thus, a polymorphism can predispose individuals to an accumulation of somatic mtDNA mutations, OXPHOS impairment, and, therefore, to an increased or decreased risk of developing AD. Different haplogroups have been identified and related to both an increased or decreased risk of AD [114]. However, these studies are controversial and deeper investigations are required to fully understand the possible role of mtDNA polymorphism in the pathogenesis of AD [115,116,117].

### 5.2. Parkinson’s Disease

Parkinson’s disease (PD) is the second most common neurodegenerative disorder in the elderly population and is pathologically defined by the loss of the neuromelanin-containing dopaminergic neurons in the substantia nigra (SN) with the development of intracytoplasmic inclusions called Lewy bodies in the surviving neurons [118]. Clinically, the disease manifests itself with tremors, rigidity, and bradykinesia accompanied by nonmotor symptoms such as sleep disorders or cognitive impairment [118]. Among the NDs, PD analysis shows the best-documented information about mitochondrial dysfunction and mtDNA damage [119,120]. Familial PD is characterised by mutations in proteins involved in mitochondrial pathways that elicit oxidative stress and are linked to increased mtDNA damage [121]. Sporadic PD patients, representing 80% of the cases, show an association with mitochondrial dysfunction [122]. Studies on post mortem PD brain tissue revealed a high level of apurinic/apyrimidinic (abasic) sites in mtDNA of nigral neurons but not in cortical ones [123].

Studies deploying the respiratory chain complex I inhibitor rotenone in a rat PD model demonstrated that mtDNA damage is detectable before the onset of neurodegeneration and is produced selectively only in the midbrain neurons, suggesting that mtDNA can be considered an early marker of PD. Indeed, when complex I was inhibited, midbrain neurons produced more H_2_O_2_ compared to cortical neurons, consequently resulting in oxidative damage on their mtDNA [123]. The presence of abasic sites may also underline an impairment of the BER pathway in mitochondria of PD individuals. To this end, Davidzon et al. showed that POLγ mutations can cause early-onset Parkinsonism and are related to multiple mtDNA deletions in muscle [128]. However, more recently, Dai et al. reported that in POLγ mutant mice, besides the increased mtDNA deletions in SN dopaminergic neurons, no signs of mitochondrial dysfunction or degeneration were found [129]. Further studies are necessary to better elucidate the involvement of POLγ in the onset of PD.

It has also been observed that there is an accumulation of large deletions in mtDNA and an absence of point mutations in PD patients [130,131,132]. In addition, it has been hypothesised that a failure in detecting point mutations in late-stage PD tissues might be caused by the degeneration of neurons carrying these mutations at earlier stages of the disease [133].

### 5.3. Amyotrophic Lateral Sclerosis

Amyotrophic lateral sclerosis (ALS) is a neurodegenerative disorder occurring during adulthood and characterised by a progressive degeneration of motor neurons. In about 75–80% of the patients, symptoms manifest first in either the upper or lower limbs, while the remaining 20% of patients develop bulbar symptoms such as dysarthria or dysphagia [134]. The first autosomal dominant mutation associated with familiar ALS was identified in the *SOD1* gene codifying for the cytosolic copper-zinc superoxide dismutase protein [135]. Since SOD1 is a scavenger enzyme involved in the maintenance of cellular redox balance, several investigations have focused on the possible contribution of oxidative stress and mitochondrial damage in the pathophysiology of ALS. Although there is not yet any evidence demonstrating the direct involvement of SOD1 mutations in producing mitochondrial dysfunction, it has been clearly shown that the most severe forms of ALS carry a higher frequency of mtDNA lesions [136]. Murakami et al. proposed that mutant SOD1 displays less protective activity against oxidative stress, resulting in an early and selective impairment of DNA repair enzymes rather than a direct effect on ROS [137]. Consistently, it has been demonstrated that the nDNA repair systems are altered in ALS, leading to DNA oxidation and neuronal dysfunction [125,126]. Notably, spinal motor neurons under physiological conditions express higher levels of DNA repair enzymes. Thus, they are more sensible to impairment of the DNA repair systems in ALS. In light of the results reported above, we can hypothesise that mtDNA damages due to an altered BER system are involved in spinal motor neuron degeneration.

### 5.4. Huntington’s Disease

Huntington’s disease (HD) is an autosomal dominant neurodegenerative disorder characterised by an aberrant expansion of the cytosine adenine guanine (CAG) triplet in the polyglutamine region of the huntingtin (*HTT*) gene [138]. Mutant huntingtin protein (htt) is neurotoxic, leading to the symptoms of HD such as motor dysfunction, cognitive impairment, and psychiatric disturbances. The molecular mechanisms driven by mutant htt and leading to HD are not yet fully elucidated, but evidence suggests that mutant htt affects the transcription of certain genes and promotes mitochondrial dysfunction [139,140]. Mutant htt inhibits the expression of peroxisome proliferation-activated receptor gamma coactivator 1α (PGC-1α), leading to mitochondrial dysfunction [141]. PGC-1α has a neuroprotective role in transgenic HD mice, suggesting that its dysregulation can negatively affect neuronal physiology. Impaired energy metabolism is an early marker in presymptomatic individuals, suggesting a link between mutant htt and altered mitochondrial function. Both HD mice models and HD patients show several mtDNA mutations and depletions as well as a reduced activity of mitochondrial complex II and III [142,143,144]. Finally, HD is also characterised by impaired calcium homeostasis, which is controlled by mitochondria [145].

**Table 2 ijms-23-11391-t002:** Haplogroups identified for AD and PD.

**Haplogroup AD**	**Activity**	**References**
K and U	Protective effect: neutralises the harmful effect of the APOE ε4 allele	[146]
HV	Acts synergistically with the APOE4 allele, significantly associated with the risk of AD	[116]
H	Acts synergistically with the APOE4 allele, significantly associated with the risk of AD	[147,148]
HV5	Acts synergistically with the APOE4 allele, significantly associated with the risk of AD	[149]
B5a	Genetic susceptibility to AD	[150]
L1 and L3	L1 participants were at a significantly increased risk for developing dementia. L3 participants exhibited higher Aβ42 levels	[151]
UK, T	Disparity between studies and no congruent data	[152]
UK, K, J, and JT	No validated relevance and no congruent studies	[153]
**Haplogroup PD**	**Activity**	**References**
A5	PD-promoting	[154]
B5	Preventive, resistance against PD	[154,155]
UKJT and R	Protective: 22% reduction in population-attributable risk for PD	[156,157]
J and K	Protective: decreased PD risk	[158,159,160]
D	Increased PD risk	[161]
B	Decreased PD risk	[161]

## 6. Conclusions

The studies analysing the role of mtDNA damage in promoting the onset and progression of neurodegenerative diseases are still controversial because they are very recent. More investigations have been carried out in PD and ALS, where the familiar forms of these pathologies are clearly linked to alterations in genes directly involving mitochondrial function. However, the majority of patients affected by NDs such as AD, PD, and ALS present a “sporadic” form, which is not directly linked to mutations in those genes promoting familiar NDs. Notably, these sporadic NDs are all characterised by impaired energy metabolism, suggesting that alterations in mitochondrial function play a key pathophysiological role in these diseases. Although very preliminary, the results analysing the presence of defined mtDNA haplogroups are very relevant, because they may provide an explanation of the incidence of NDs in defined populations or families that do not carry any mutation in the genes responsible for the familiar forms of HDs. Knowing the alterations in mtDNA is important in order to investigate the mechanistic pathways triggered by the environmental and metabolic alterations that are known to be risk factors for sporadic NDs. More studies are necessary to define the role of mtDNA damage in the onset of sporadic NDs. Nevertheless, the current results confirm that analysis of such mtDNA alterations can pave the way for a better understanding of the molecular mechanism involved in sporadic NDs.

## Figures and Tables

**Figure 1 ijms-23-11391-f001:**
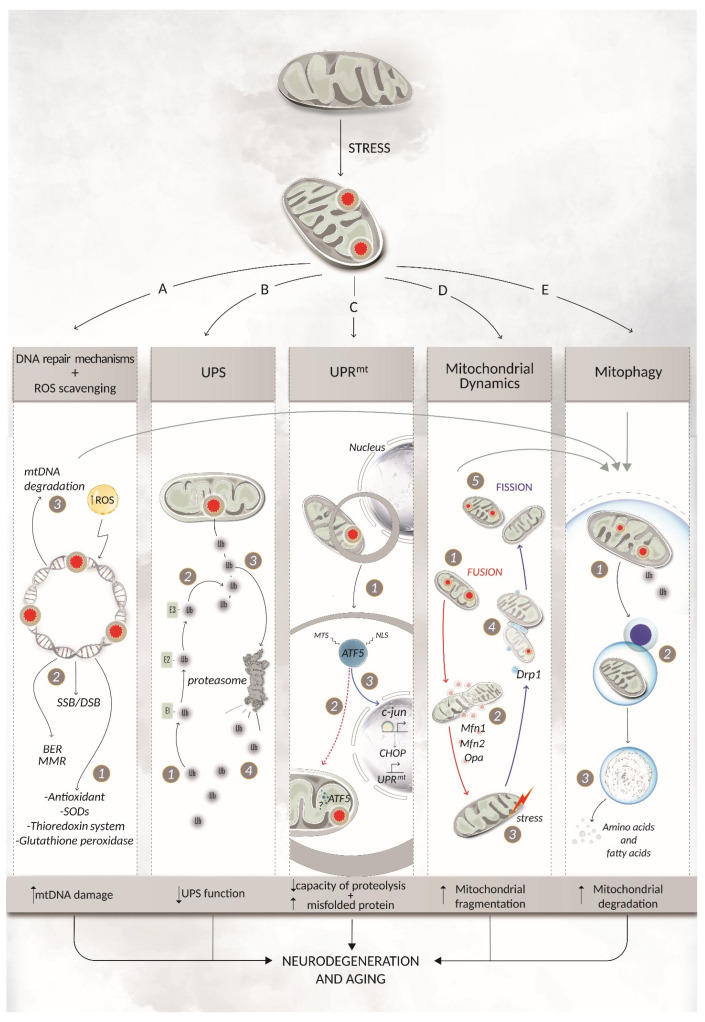
Mitochondrial quality control: regulating mitochondrial turnover and homeostatic mtDNA repair mechanisms and ROS scavenging (A). Various mechanisms can be activated to either prevent or eliminate mtDNA damage. The first line of defence involves ROS scavenging factors such as antioxidants, SOD, the thioredoxin system, and glutathione peroxidase (1). Repair mechanisms become activated after damage is sensed on the mtDNA. Among these, the mitochondrial BER pathway is the most well-characterised, but in the last decades, other common proteins of nDNA repair mechanisms have also been identified in mitochondria, shedding light on the possibility that these pathways have a role in mtDNA repair as well (2) [13,14]. If all these damage responses are not able to fully repair the lesion, then a single molecule of mtDNA can be degraded. This does not impact organelle physiology since each mitochondrion owns multiple copies of the same nucleic acid (3). Finally, if the damage is extensive, the whole mitochondrion can be degraded through mitophagy. Ubiquitin proteasome system (UPS) (B). Dysfunctional mitochondrial proteins can be degraded by the UPS, a specific degradation system which relies on the covalent binding of ubiquitin to lysine residues within target proteins. Ubiquitin is translocated via the E1, E2, and E3 enzymes (1) before reaching the damaged protein (2). The polyubiquitin-tagged protein is then translocated to the cytosolic proteasome for degradation (3), where the ubiquitin is recycled, ready for another round of polyubiquitination (4). UPS is crucial in preserving mitochondrial integrity and vice versa. Indeed, dysfunctional mitochondria with an increased number of damaged proteins could not only overflow the proteasome but also affect the proteasomal subunits themselves, thereby affecting the catalytic activity of the UPS. Once mitochondrial dysfunction and proteasomal impairments develop, a vicious cycle may start, leading to a progressive failure of the UPS and, consequently, to ageing or, in the worst scenario, to neurodegenerative diseases. Mitochondrial unfolded protein response (UPR^mt^) (C). The UPR^mt^ system can be activated in response to an accumulation of unfolded proteins in mitochondria. The crucial role of the UPR^mt^ protein ATF5 is explicated through its nuclear localisation sequence (NLS) and mitochondria targeting sequence (MTS) (1). Under physiological conditions, ATF5 is localised in mitochondria (red dashed arrow) and likely degraded by a protease (2) such as the one characterised in *C. elegans* [15]. If mitochondrial import is dysfunctional, ATF5 accumulates in the cytosol and is translocated into the nucleus (blue arrow), where it can influence the activation of the transcriptional factors c-Jun and CHOP, which in turn regulate the activation of genes able to restore mitochondrial functions (3). Mitochondrial dynamics (D). Mitochondria can orchestrate cycles of fusion and fission as part of their dynamic network, allowing the maintenance of shape, distribution, and size. This mechanism can also be used to cope with unrepairable damages such as inter-/intrastrand and DNA-protein cross-links through the removal of the damaged section of the mtDNA by mitophagy [16]. Fusion ((1), red arrows) is mediated by Mfn1, Mfn2, and Opa1 (2) and allows the mitochondria to bond together to respond to damage (3). Fission (blue arrows) can also mediate the response to an external stress (3) that causes mitochondrial dysfunction. Fission is mediated by dynamin-related protein 1 (Drp1) (4) and, opposite to fusion, acts to isolate the damaged area of the organelle for clearance by mitophagy (5). Mitophagy (E). Defective mitochondria can be cleared in a process called mitophagy. The whole organelle is isolated from the rest of the cell owing to the generation of an autophagosome (1). The fusion of the autophagosome with a lysosome gives rise to the autolysosome (2) containing a set of enzymes in an acidic environment which drives the degradation of proteins, lipids, and nucleic acids in a controlled manner (3). BER = base excision repair; DSB = double strand break; MMR = mismatch repair; ROS = reactive oxygen species; SOD = super oxidase dismutase; Ub = ubiquitin; UPR^mt^ = mitochondrial unfolded protein response; UPS = ubiquitin proteasome system.

**Figure 2 ijms-23-11391-f002:**
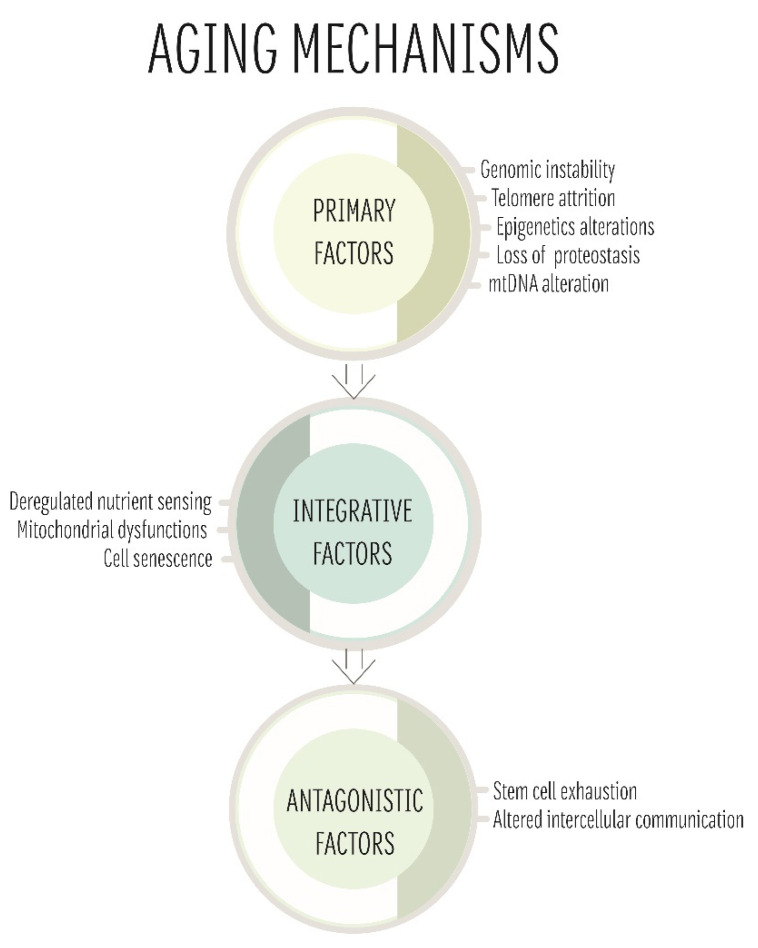
Ageing mechanisms.

**Figure 3 ijms-23-11391-f003:**
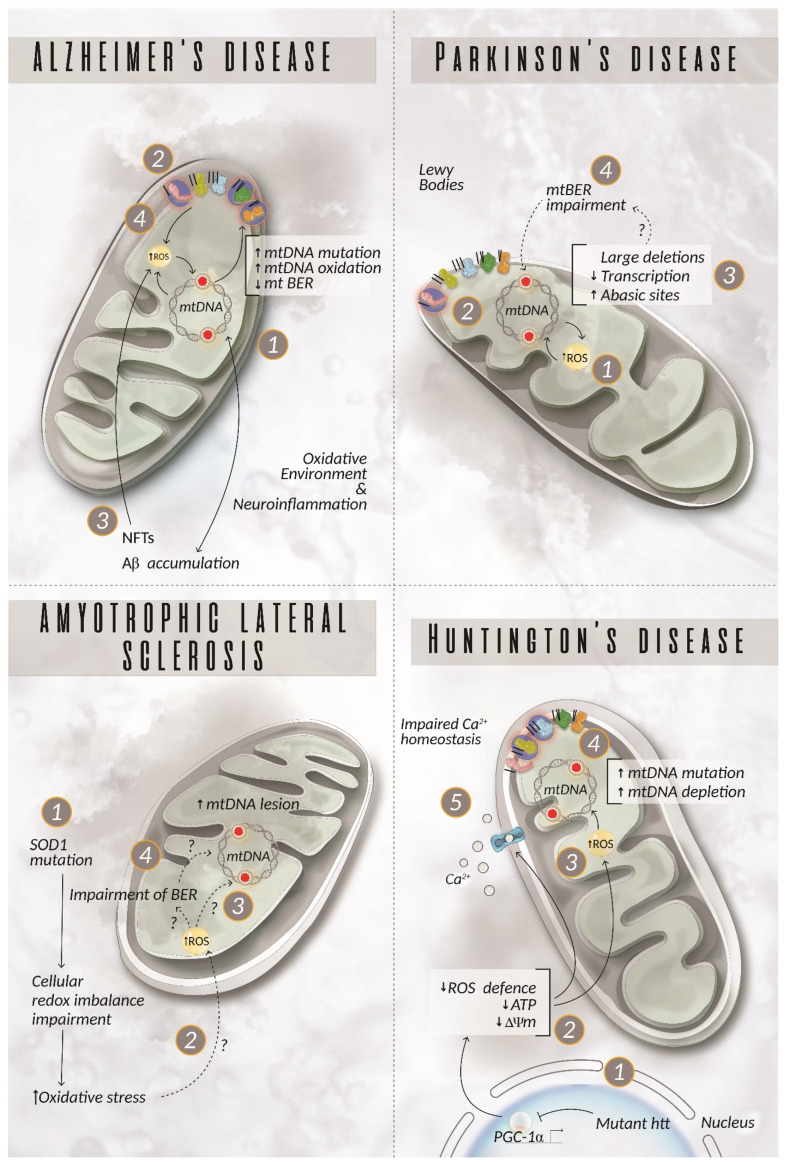
mtDNA damage in neurodegenerative diseases. Alzheimer’s disease (AD). mtDNA damage in AD can lead to energy failure (1) driven by the defective complexes I, III, IV, and V (2), promotion of Aβ accumulation (3) and increased oxidative stress (4), which, in turn, exacerbates mtDNA damage and increased production of ROS, creating a vicious cycle of dysfunctional and damaged mitochondria. Parkinson’s disease (PD). An incremented production of ROS (1) increases the susceptibility of mtDNA to damage (2). PD patients display either an increased amount of abasic sites and/or large mtDNA deletions (3) with a consequent failure in the formation of a fully functional OXPHOS system. Up to now, there have been no clear explanations about the mechanisms underpinning the dysfunction detected in the mitochondria in patient neurons, but the involvement of defects at the level of the mitochondrial BER is plausible (4). Amyotrophic lateral sclerosis (ALS). Mutations in SOD1 cause a cellular redox imbalance (1), but it is not clear how this phenomenon affects mtDNA stability. The most recent theory suggests an indirect role of mutated SOD1 on the proteins involved in mtDNA repair rather than a direct effect on ROS production with consequent mtDNA damage (2 and 3) [124]. As of yet, there is no evidence for this hypothesis even though the alteration of the nDNA repair system supports the impairment of the mitochondrial BER as documented in ALS (4) [125,126,127]. Huntington’s disease (HD). Mutant huntingtin (htt) has indirect toxic effects on mtDNA. It suppresses the expression of PGC-1α (1), which negatively impacts ROS scavenging mechanisms, ATP production, mitochondrial membrane potential, and, more generally, the whole mitochondrial physiology (2). Correspondingly, ROS production is exacerbated (3), leading to increased mtDNA mutation and depletion, consequently disrupting mitochondrial integrity (4). Studies have also underlined the presence of imbalanced Ca^2+^ homeostasis in HD patients (5). Highlighted complexes with a circle represent components of the OXPHOS system, in which a mutation related to the disease described has been reported. Aβ = amyloid beta peptide; BER = base excision repair; NFTs = neurofibrillary tangles; ROS = reactive oxygen species; SOD1 = superoxide dismutase 1.

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
