# Peer review of "Mitochondrial DNA Repair in Neurodegenerative Diseases and Ageing"

_ijms, 2022, doi:10.3390/ijms231911391_

Round 1

Reviewer 1 Report

The manuscript “Mitochondrial DNA repair in neurodegenerative diseases and ageing” by Bazzani et al. reviews the effect of mtDNA homeostasis failure in promoting ageing and neurodegenerative diseases such as Alzheimer’s or Parkinson’s disease among others. Authors summarize the most common types of mtDNA alterations and describe the different aspects of mitochondria quality control such as DNA repair mechanisms, ROS scavenging or mitochondrial dynamics among others. Finally, they focus on the physiological role of mtDNA lesions in neurogenerative diseases.

The review was well conducted and the results are clearly presented and English language and style are also fine. The conclusions are supported by bibliography but I some comments:

-        Line 75: superoxide should be written O2- instead of O2-

-        Line 76: “…during oxidative phosphorylation system” should be “… oxidative phosphorylation process” or “… oxidative phosphorylation”

-        Line 184: there is an 8 written in the word there

-        Line 453: Is complex III not missing?

-        Lines 468 and 523: PCG-1α should be PGC-1α

-        Tables 1 and 2 should be cited in the text.

Reviewer 2 Report

There seem to be way too many minor mistakes and, in this respect, the review manuscript deserves a complete overhaul for one to be able to comprehend the full storyline.

1) Please change "Kbp" to "kbp" (line 12).

2) Please replace "double stranded" with "double-stranded" (line 12).

3) Please change "inner membrane" to either "mitochondrial inner membrane" or "inner mitochondrial membrane" (line 14).

4) Please replace "lesions, such" with "lesions such" (line 17).

5) Please change "focus" to "focuses" (line 23).

6) Please replace "neurodegenerative diseases" with "neurodegenerative disorders" (line 24).

7) Please change "Alzheimer disease; Parkinson disease" to "Alzheimer's disease; Parkinson's disease" (line 27).

8) Please replace "functions, including" with "functions including" (line 31).

9) It is not clear what the authors are concretely referring to as "subsequent signalling cascade" in "Their function is crucial for ATP production, synthesis of Heme and steroid hormones, calcium and iron homeostasis and subsequent signalling cascade [1], fatty acid oxidation, and the regulation of inter-organelles contact with the endoplasmic reticulum (ER)" (line 32)?

10) Please change "Heme" to "heme" (line 33).

11) Please replace "inter-organelles" with "inter-organelle" (line 34).

12) Please replace "mitochondria" with "mitochondrial" (lines 35, 38, 249, 257, 258, 275, 447, 470, 504).

13) Please change "aimed to rapidly respond" to something like "rapidly responding" (line 36).

14) Please replace "unfunctional" with "dysfunctional" (line 40).

15) Please change "in" to "of" (line 42).

16) Please change "mitochondria-specific, unfolded" to "mitochondria-specific unfolded" (line 43).

17) Please replace "[6] (Figure 1)" with "(Figure 1) [6]" (line 45).

18) Please replace "exerting" with "exerts" (line 46).

19) Please change "demands" to "demand" (line 47).

20) Please change "oxidative phosphorylation" to "oxidative phosphorylation (OXPHOS)" (line 48), "oxidative phosphorylation system (OXPHOS)" to "OXPHOS system" (line 75), and "oxidative phosphorylation process" to "OXPHOS" (line 255).

21) Please change "mass, but" to "mass but" (line 49).

22) Please replace "1m" with "1 m" (line 50).

23) Please change "reveal" to "have revealed" (line 53).

24) Please replace "precede" with "precedes" (line 54).

25) From "In this contest, mitochondrial alterations promote key events occurring in ageing and in neurodegeneration. In this review we focus on the crucial role of mtDNA maintenance/alterations in promoting neurodegeneration" (line 60) is not clear whether the authors are referring to "mtDNA maintenance and alterations" or "mtDNA maintenance or alterations" (line 60)?

26) Please change "mt DNA" to "mtDNA" (line 65).

27) Please replace "ribonucleotides" with "ribonucleotide" (line 72).

28) Please replace "Antioxidants" with "Antioxidant" (line 81).

29) Please change "scavengers" to "scavenger" (line 81).

30) Please replace "as the" with "as" (line 81).

31) Please change "and the" to "and" (lines 82, 93).

32) Please change "counteract the" to "counteract" (line 83).

33) Please replace "preserving the" with something like "thus preserving" (line 83).

34) Please change "produced by" to "associated with" (line 83).

35) Please replace "proinflammatory cascade" with either "the proinflammatory cascade" or "proinflammatory cascades" (line 86).

36) Please provide an upward arrow infront of the "Mitochondrial fragmentation" caption in Figure 1 (Mitochondrial Dynamics).

37) Please change "sistem" to "system" in Figure 1 (DNA repair mechanisms + ROS scavenging).

38) Both the red dashed (UPRmt), blue (UPRmt), orange (Mitochondrial Dynamics), blue (Mitochondrial Dynamics) arrows indicated in Figure 1 are hardly visible. Please fix.

39) The role of "C-JUN", "stress", "LC3", and "Aa+ fatty acids" in the processes depicted in Figure 1 is rather obscure as there is no explicit description provided in the respective figure legend. Please fix. In addition, what does "Aa+" refer to?

40) Please replace "Mitochondria" with "Mitochondrial" (line 90).

41) Please change "homeostasis" to "homeostatic" (line 90).

42) Please replace "Base Excision Repair" with "base excision repair" (lines 95).

43) Please change "nuclear DNA" to "nDNA" (line 96, 341, 466, 508).

44) Please change "throwing" to "shedding" (line 97).

45) Please change "too" to "as well" (line 97).

46) Please replace "in the" with "in" (lines 97, 141).

47) It is not exactly clear what the authors mean by "are not enough for repair" in "If all these responses to the damage are not enough for repair, then a molecule of mtDNA can be degraded without impacting the mitochondrial physiology, thanks to the presence of multiple copies of the same nucleic acid within mitochondria" (line 98)?

48) Please change "impacting the" to "impacting" (line 99).

49) Please replace "thanks" with "owing" (lines 99, 129).

50) Please change "extended" to "extended," (line 100).

51) Please replace "Proteasome System" with "proteasome system" (line 101).

52) Please change "Ubiquitin Proteosome System (UPS)" to "UPS" (line 102).

53) Please replace "polyubiquitin tagged" with "polyubiquitin-tagged" (line 105).

54) It is not exactly clear which enzyme "catalytic activity" are the authors referring to in "Indeed, dysfunctional mitochondria with increased damaged proteins could not only overflow the proteosome, but also affect the proteasomal subunits themselves, thereby affecting the catalytic activity" (line 107)?

55) Please change "increased" to "accumulated" (line 108).

56) Please replace "proteosome, but" with "proteosome but" (line 108).

57) It is not clear what the authors mean by "both system" in "Once dysfunction and proteasomal impairments develops then a vicious cycle may start, leading to a progressive failure of both system and consequently to ageing or, in the worst scenario, to neurodegenerative diseases" (line 109)?

58) Please change "system" to "systems" (line 111).

59) Please replace "(C)" with "(C)." (line 112).

60) Please change "to an" to "to" (line 113).

61) Please replace "proteins detected" with "proteins" (line 113).

62) Please change "Mitochondria Targeting Sequence" to "mitochondria targeting sequence" (line 114).

63) It is not exactly clear what the authors mean by "better characterized" in "In physiological conditions ATF5 is localized in mitochondria (red dashed arrow) and likely degraded by a protease (2) as better characterized in C. elegans" (line 115)? Better than what?

64) Please replace "In physiological conditions" with "Under physiological conditions," (line 115).

65) Please change "cytosol, and it" to "cytosol and" (line 117).

66) Please change "in" to "into" (line 118).

67) Please replace "(3)" with "(3)." (line 119).

68) Please change "(D)" to "(D)." (line 120).

69) Please replace "distribution, and" with "distribution and" (line 121).

70) It is not exactly clear what the authors mean by "inter-intra-strand" in "This mechanism can also be used to cope with unrepairable damages as inter-intra-strand and DNA-proteins cross-links, through the removal of the damaged part through mitophagy" (line 121)?

71) Please change "damages" to "damages such" (line 122).

72) Please replace "cross-links, through" with "cross-links through" (line 122).

73) Please change "through mitophagy" to "by mitophagy" (line 123).

74) Please replace "through" with "by" (lines 124, 126, 127).

75) Please change "(3) which cause" to "(3), which causes" (line 125).

76) Please replace "(E)" with "(E)." (line 128).

77) Please change "originates" to "gives rise to" (line 130).

78) Please replace "drives" with "drive" (line 131).

79) Please change "lipids, and" to "lipids and" (line 131).

80) Please replace "procedure" with either "fashion" or "manner" (line 132).

81) Please change "Base Excision Repair (pathway)" with "base excision repair pathway" (lines 95).

82) Please change "Double Strand Break Repair (pathway)" to "double-strand break repair pathway" (line 133).

83) Please replace "mismatch Repair (pathway)" with "mismatch repair pathway" (line 133).

84) Please change "Reactive Oxygen Species" to "reactive oxygen species" (line 133).

85) Please replace "Super Oxidase Dismutase" with "super oxidase dismutase" (line 134).

86) Please change "Ubiquitin" to "ubiquitin" (line 134).

87) Please replace "Unfolded Protein Control" with "unfolded protein control" (line 134).

88) Please change "Ubiquitin Proteasome System" to "ubiquitin proteasome system" (line 135).

89) Please change "alter the" to "alter" (line 136).

90) Please replace "G->A/ T->C" with "G->A/T->C" (line 143).

91) Please change "polymerase " to "polymerase (POL)" (line 149), "polymerase " to "POL" (line 190), and "polymerase (POL)" to "POL" (line 483).

92) It is not clear what the authors mean by "major arc" in "The majority of the mtDNA deletions occur in the major arc of the mtDNA and have been associated to different pathologies, where the clinical prognosis directly correlates with the mtDNA deletion frequency" (line 157)?

93) Please change "to" to "with" (lines 158, 498).

94) It is not clear what the authors mean by "minor arc" in "Nonetheless, specific deletions are found also on the minor arc" (line 160)?

95) Please replace "pathologies -" with "pathologies –" (line 162).

96) Please change "Single Strand breaks and Double Strand Breaks" to "Single-strand and double-strand breaks" (line 166).

97) From "Single strand breaks (SSBs) and double strand breaks (DSBs) are discontinuities in one/two strands of the DNA that can occur on mitochondrial DNA directly (e.g., from attack of reactive oxygen species) or indirectly (e.g., during the enzymatic cleavage of the phosphodiester backbone performed throughout the base excision repair pathway)" (line 167) is not clear whather the authors are referring to "one and two" or "one or two" strands of the DNA?

98) Please replace "Single strand" with "Single-strand" (line 167).

99) Please change "double strand" to "double-strand" (lines 167, 285).

100) Please replace "mitochondrial DNA" with "mtDNA" (lines 168, 484).

101) Please change "reactive oxygen species" to "ROS" (line 169).

102) Please replace "performed throughout the base excision repair" with "mediated by the BER" (line 170).

103) Please change "mechanism of clearance" to "clearance mechanism" (line 172).

104) Please replace "mtDNA" with "Mitochondrial DNA" (line 173).

105) Please change "Base Excision Repair (BER)" to "BER" (line 176).

106) Please replace "(the same function can be performed by the glycosylase itself)" with "(the same function can be performed by the glycosylase itself)." (line 182).

107) Please incorporate "Step (2)" so that it becomes part of a sentence (line 183).

108) Please change "t8here" to "there" (line 184).

109) Please replace "glycosylases:" with "glycosylases," (line 184).

110) Please change "create the" to "create a" (line 186).

111) Please replace "alterations: alkylation" with "alterations. Alkylation" (line 188).

112) Please change "glycosylases" to "glycosylase" (line 189).

113) Please replace "This process is called short-patch BER (SP-BER). When" with "When" (line 191).

114) Please change "sub-pathways: the short-patch" to "sub-pathways, the short-patch (SP-BER)" (line 192).

115) Please replace "process: in the SP-BER" with "process. In the SP-BER," (line 195).

116) Please change "involved in the process" to "involved" (line 199).

117) Please replace "mitochondria" with "mitochondria," (lines 199, 214).

118) Please change "described (DNA ligase III)" to "described, DNA ligase III" (line 200).

119) Please replace "in nuclei also present is the DNA ligase I" with "DNA ligase I is also present in nuclei" (line 201).

120) Please merge "More interesting is the study of LP-BER sub-pathway in mitochondria" within the same paragraph (line 202).

121) Please change "recently" to "recently," (line 202).

122) Please replace "within mitochondria only the SP-BER is active" with "only SP-BER is active within mitochondria " (line 203).

123) Please change "10 years" to "decade" (line 204).

124) Please replace "Double strand" to "double-strand" (line 206).

125) Please change "mismatch repair pathway (MMR)" to "mismatch repair (MMR) pathway" (line 207).

126) Please replace "have" with "has" (line 208).

127) Please replace "mismatch repair" with "MMR" (line 209).

128) Please provide references for "Indeed, recent studies demonstrated that YB-1 interacts with the glycosylase NEIL2, APE1 and the DNA ligase III" (line 210).

129) Please change "studies are" to "research is" (line 211).

130) Please replace "double strand breaks (DSBs)" with "DSBs" (line 213).

131) The meaning of "other reports support a not-well characterised microhomology-mediated end-joining (MMEJ) repair pathway [45-47] rather than the NHEJ" is rather vague in "Interestingly, other reports support a not-well characterised microhomology-mediated end-joining (MMEJ) repair pathway [45-47] rather than the NHEJ, which appear to be undetectable in mitochondria" (line 220) as it is not explicitly clear whether this relates to nuclear or mitochondrial DNA repair.

132) Please change "appear" to "appears" (line 222).

133) It is not exactly clear what the authors mean by "cases" in "This hypothesis is supported by the detection of short repetitive sequences flanking the deletions in mtDNA occurring in 85% of the cases [48, 49] and suggesting the existence of a recombination mechanism involved in the maintenance of the mtDNA integrity" (line 222)?

134) Please replace "suggesting" with "suggests" (line 224).

135) Please replace "of the" with "of" (lines 225, 400, 425, 500).

136) Please change "performs the" to "undergoes" (line 231).

137) Please replace "cell" with "cell," (line 236).

138) Please change "behind the" to "underlying" (line 239).

139) From "Two mechanisms have been proposed: (i) degradation by nucleases [53] and (ii) elimination of the whole mitochondria carrying the lesion via autophagy/mitophagy" (line 240) is not clear whether the authors are referring to "autophagy and mitophagy" or "autophagy or mitophagy"?

140) "seems" appears twice in "It seems that the mtDNA degradation is activated in presence of excess DNA damage. This mechanism is not damage-specific, and its kinetics varies depending on the cell lines. Recently it has been shown that linear DNA formed upon DSB seems to be degraded via the exonuclease activity of pol and MGME1" (line 242). Please fix.

141) Please replace "that the" with "that" (lines 242, 317).

142) Please replace "in" with "in the" (line 242).

143) Please change "damage-specific, and" to "damage-specific and" (line 243).

144) It is not clear what "cell lines" are the authors referring to in "This mechanism is not damage-specific, and its kinetics varies depending on the cell lines" (line 243) as there seems to be no contextual mention of cell cultures?

145) Please replace "Recently" with "Recently," (line 244).

146) Please change "DSB" to "DSBs" (line 244).

147) Please replace "It is still debated the role of defined enzymes in this pathway, such as Twinkle helicase [57] and mtSSB [58], which can assist particularly MGME1 during the degradation" (line 245) with something like "Nevertheless, the role of other mtDNA replication enzymes such as Twinkle helicase [57] and mtSSB [58] in this pathway is still debated".

148) Please replace "2020" with "2020," (line 248).

149) Please change "of" to "on" (line 250).

150) Please replace "behind the initiation of mitophagy" with "of mitophagy initiation" (line 252).

151) Please change "the cells" to "cells" (line 253).

152) "Overall, it has been hypothesised that mitophagy occurs after irreparable mtDNA damages, which stimulate mitochondria fission followed by mitophagy of dysfunctional mitochondria daughter" (line 256) does not make sense due to the following reasons:

a) that mitophagy proceeds after mtDNA damage/mitochondrial fission is redundantly mentioned twice

b) it is not clear what the authors mean by "mitochondria daughter"?

c) the reason why only one "mitochondria daughter" is subject to mitophagy is not explained

153) Please change "occurs after" to "is triggered by" (line 256).

154) Please replace "Fission" with "Mitochondrial fission" (lines 262, 266).

155) Please change "Dynamin relate protein I (DrpI)" to "dynamin-related protein 1 (Drp1)" (line 263).

156) Please replace "outer membrane" with "mitochondrial outer membrane" or "outer mitochondrial membrane" (line 264).

157) Please change "Fusion" to "Mitochondrial fusion" (line 264).

158) Please replace "Mitofusin" with "mitofusin" (line 265).

159) From "Fission and fusion are critical to cope with damaged DNA" is not clear whether the authors are referring to mitochondrial or nuclear DNA (line 266)?

160) Please change "critical" to something like "critical for the ability of cell" (line 266).

161) "Bulky adducts that arise on the mtDNA cannot be repaired by pathway as the nucleotide excision repair (NER) pathway as happens in the nucleus" (line 266) does not seem to be grammatically correct with respect to "by pathway as the nucleotide excision repair (NER) pathway as happens". Please rephrase.

162) Please replace "on the" with "on" (line 267).

163) Please change "happens" to "occurs" (line 268).

164) "Indeed, it seems that mtDNA damages such as pyramidine dimers, base modifications or inter- intra-strand cross-links and DNA-protein cross-links are cleared by degradation, fission, fusion and mitophagy" (line 269) is confusing as it claims that mtDNA damage can be cleared by both mitochondrial fission and fusion. Please either explain how is it possible that two opposing processes lead to the same outcome or rephrase the sentence accordingly.

165) It is not exactly clear what the authors mean by "inter- intra-strand cross-links" and "degradation" in "Indeed, it seems that mtDNA damages such as pyramidine dimers, base modifications or inter- intra-strand cross-links and DNA-protein cross-links are cleared by degradation, fission, fusion and mitophagy" (line 269)?

166) Please replace "it seems that mtDNA" with "mtDNA" (line 269).

167) Please change "pyramidine" to "pyrymidine" (line 270).

168) Please replace "intra-strand cross-links" with "intra-strand" (line 270).

169) Please change "fission, fusion and" to "mitochondrial fission or fusion or by" (line 271).

170) From "Moreover, it is still unclear whether mtDNA damage and fission/fusion dynamics are correlative or consequential" (line 272) is not clear whether the authors are referring to "fission and fusion" or "fission or fusion"?

171) Please replace "it is still unclear whether mtDNA damage and fission/fusion dynamics are correlative or consequential" with "whether mtDNA damage and mitochondrial fission/fusion dynamics are correlative or consequential is also elusive" (line 272).

172) From "On the other hand, altered Drp1 activity and its recruitment to mitochondria participate in neurodegeneration by promoting mitochondria dysfunction" (line 273) is not clear whether the authors are referring to increased or decreased Drp1 activity?

173) Please change "participate in" to "contributes to" (line 274).

174) Please replace "the base excision repair (BER) pathway" with "BER" (line 278).

175) Please change "the proteins of the BER in the" to something like "the BER machinery in" (line 279).

176) Please replace "difference with the" with "differences to its" (line 280).

177) From "Here, we reported a list of proteins known to be involved in different nuclear repair pathways and that have been studied for their role in mitochondrial integrity and stability" (line 281) is not entirely clear whether the authors are referring to "mitochondrial integrity and stability" as the whole organelle or just mtDNA?

178) Please change "reported" to "report" (line 281).

179) Please replace "present in the mitochondria alone" with "with exclusive mitochondrial localization" (line 283).

180) It is not exactly clear what the authors mean by "Potentially involved in degradation" in Table 1 (Twinkle)? Degradation of what?

181) Please change "Remove" to "Removal of" (ExoG), "DNA binding" to "DNA-binding" (YB-1), "mammals’" to "mammalian" (HR/NHEJ), "Unwind" to "Unwinding of" (Twinkle), "Enhance" to "Enhancing" (mtSSB) in Table 1

182) Please sort the abbreviation list of the legend to Table 1 in an alphabetical order.

183) Please replace "non homologous" with "non-homologous" (line 285).

184) Please change "single strand" to "single-strand" (line 285).

185) Please replace "damage, such" with "damage such" (line 290).

186) Please change "[76] (Figure 2)" to "(Figure 2) [76]" (line 292).

187) Please merge "The focal point of this section of the review is to give an overview of the role of mitochondria and the increasing interest in this organelle in the ageing processes" (line 293) within the same paragraph.

188) Please replace "section of the review" with "section" (line 293).

189) Please change "here summarised" to "summarised here" (line 295).

190) It is not clear what the authors mean by "All these approaches" in "All these approaches show both limitations and strengths and a comprehensive theory explaining mitochondrial role in ageing is still missing" (line 295) as there is no prior mention of approaches?

191) Please replace "explaining" with "explaining the" (line 296).

192) Please change "this contest" to "in the context of ageing" (line 298).

193) Please replace "Free Radical Theory of Ageing" with "free radical theory of ageing" (line 307).

194) Please change "Mitochondrial Free Radical Theory of Ageing" to "mitochondrial free radical theory of ageing" (line 308).

195) Please replace "Dr. Harman’s" with "Harman’s" (line 310).

196) Please replace "damage, through" with "damage through" (line 311).

197) Please change "with" to "to" (line 312).

198) Please replace "Dr. Harman" with "Harman" (line 314).

199) Please change "is caused" to "originates" (line 317).

200) Please replace "2005" with "2005," (line 320).

201) Please change "cells, such" to "cells such" (line 320).

202) Please replace "cells, but" with "cells but" (line 321).

203) Please format "a" consistently with the rest of the text (line 323).

204) Please change "a destroyed respiratory capability, and" to "impaired respiratory capacity and" (line 323).

205) Please replace "of this theory are" with "for this theory is" (line 324).

206) Please change "on" to "of" (line 326).

207) Please replace "Sod2" with "SOD2" (line 326).

208) Please change "Grey`s" to "Grey's" (line 328).

209) It is not exactly clear what "mortality" are the author referring to in "In addition, other studies show that increased oxidative stress, induced by glucose restriction or by hypoxia, is beneficial due to the promotion of cellular resistance to stress, resulting in reduced mortality" (line 329)? Mortality of patients, laboratory model organisms, or cells?

210) Please replace "the promotion" with "activation" (line 330).

211) From "All together, these data suggest that mtDNA mutations/alteration promote ageing in an independent manner to oxidative stress" (line 331) is not clear whether the authors are referring to mtDNA "mutations and alteration" or "mutations or alteration"?

212) Please change "Expansion Theory" to "expansion theory" (line 333).

213) Please replace "In 1988, a link between mtDNA deleterious mutations and ageing was first suggested" with "A link between mtDNA deleterious mutations and ageing was first suggested in 1988" (line 334).

214) Please change "within mtDNA accumulate" to "accumulate within mtDNA" (line 336).

215) "specific" appears twice in "Clonal expansion is a process well characterised in B-lymphocytes, whereby a large number of cells can be selected for and produced with a specific genotype characterised by specific DNA deletions" (line 337). Please rephrase.

216) Please replace "and produced with" with "yielding" (line 338).

217) Please change "ageing, called “Clonal Expansion Theory”" to "ageing called “clonal expansion theory”" (line 340).

218) Please replace "nuclear DNA mutations promoting cancers" with "cancer-promoting nDNA mutations" (line 341).

219) Please change "mutations on mtDNA" to "mtDNA mutations" (line 342).

220) Please replace "up, rather" with "up rather" (line 343).

221) Please change "individuals, supports" to "individuals supports" (line 345).

222) "Specific point mutations and deletions in mtDNA are present from young age, already during the early development and even in germline cells" (line 345) is puzzling since it is difficult to narrowly distinguish between "from young age" and "during the early development" making these prepositions redundant to each other. In addition, are the authors referring to human aging?

223) Please replace "during the" with "during" (line 346).

224) The description of why "mutations promote premature ageing" in "These mutations promote premature ageing because of mitochondrial oxidative phosphorylation deficiency, which is a hallmark of ageing and induces metabolic alterations in proliferating cells, while decreases apoptosis" (line 349) is not clear as it is too dense and goes "backwards". Please reverse the order of processes, starting from oxidative phosphorylation, depicting the metabolic and antiapoptotic effects it has on proliferating cells, how mtDNA mutations lead to OXPHOS deficiency and what are the implications for premature aging. For increased comprehension, highlight the link between apoptosis and age-related disease. Such explanation can span one or several new sentences.

225) "These mutations"/"these mtDNA mutations" is mentioned twice in "These mutations promote premature ageing because of mitochondrial oxidative phosphorylation deficiency, which is a hallmark of ageing and induces metabolic alterations in proliferating cells, while decreases apoptosis [95]. Notably, these mtDNA mutations promote cancer in proliferating cells, while induce senescence in non-proliferating cells" (line 349). Please fix.

226) Please change "induce" to "inducing" (line 352).

227) Please replace "ROS Response Theory" with "ROS response theory" (line 354).

228) Please change "homeostasis, caused" to "homeostasis caused" (line 356).

229) Please replace "development, the organism’s" with "organismal development," (line 357).

230) Please change "to the" to "to" (line 358).

231) Please replace "in turn elevating the" with either "hence elevating" or "thereby elevating" (line 358).

232) Please change "This" to "Such" (line 364).

233) Please replace "evolution" with "evolutionary process" (line 367).

234) Please change "organisms, due" to "organisms due" (line 367).

235) Please replace "seems quite" with "has become" (line 367).

236) Please change "hard" to "difficult" (line 372).

237) Please replace "capable of describing" with "of" (line 373).

238) Please change "It seems that an" to "An" (line 374).

239) Please replace "interesting ideas" with something like "relevant concepts" (line 376).

240) In "Considering the complexity of ageing, we can hypothesise that mtDNA damage may represent one of the various mechanisms involved and can be more relevant in some organisms compared to others" (line 378) is not explicitly specified what is "one of the various mechanisms" involved in. Please fix.

241) Please change "may represent" to "represents" (line 379).

242) "the investigation of the role of mtDNA damage in ageing is extremely interesting" sounds too trivial in "However, the investigation of the role of mtDNA damage in ageing is extremely interesting, not only for the understanding of the complexity of ageing, but also to unveil the ageing-related mechanisms that participate to the progression of several diseases" (line 381). In addition, without including detailed examples, the whole sentence sounds too vague.

243) It is not exactly clear what the authors mean by "several diseases" in "However, the investigation of the role of mtDNA damage in ageing is extremely interesting, not only for the understanding of the complexity of ageing, but also to unveil the ageing-related mechanisms that participate to the progression of several diseases" (line 381) as these are nowhere specified?

244) Please replace "to" with "in" (line 383).

245) Please change "In particular, below" to "Below," (line 383).

246) Please replace "condition (such as but not exclusively, coordination or memory problems, due to initial neuronal dysfunction)" with "condition such as, but not exclusively, coordination or memory problems due to initial neuronal dysfunction" (line 390).

247) Please change "and that are consequences" to "as a consequence" (line 392).

248) Please replace "fatal, and" with "fatal and" (line 393).

249) Please change "neurodegenerative diseases" to "NDs" (line 395).

250) Please change "Amyotrophic Lateral Sclerosis" to "amyotrophic lateral sclerosis" (line 396).

251) Please change "whether the" to "whether" (line 400).

252) Please replace "NDs context" with "context of ND" (line 404).

253) Please change "5-10%" to "5–10%" (line 411).

254) Please replace "It is still unclear the impact of each of these factors on AD progression" with "The impact of each of these factors on AD progression is still unclear" (line 414).

255) Please change "to hypothesise the cooperate" to something like "that there is a synergy among these factors" (line 415).

256) Please replace "promotes A" with "A" (line 419).

257) Please specify the "ligase" mentioned in "On the other hand, various data underline the role of BER components acting in mitochondria: post-mortem brain of AD patients presents decreased levels of 5-hydroxyuracil (5OHU) incision as well as diminished ligase activity, suggesting an impaired function of the repair pathway, which can have negative consequences on the overall quality of the mtDNA" (line 421).

258) From "On the other hand, various data underline the role of BER components acting in mitochondria: post-mortem brain of AD patients presents decreased levels of 5-hydroxyuracil (5OHU) incision as well as diminished ligase activity, suggesting an impaired function of the repair pathway, which can have negative consequences on the overall quality of the mtDNA" (line 421) is not explicitly clear whether the authors are referring to nuclear or mitochondrial "repair pathway"?

259) Please change "mitochondria: post-mortem" to "Mitochondria. Post-mortem" (line 422).

260) Please replace "it has been revealed an interesting association" with "an interesting association has been revealed" (line 426).

261) From "Small differences in the encoded proteins can slightly affects the OXPHOS activity, leading to an overproduction of free radicals or to a reduction" (line 427) is not unequivocally evident what reduction are they referring to? If this is a reduction in free radicals, why is reduction mentioned alongside overproduction of the same species?

262) Please change "affects" to "affect" (line 427).

263) Please replace "developing the" with "developing" (line 430).

264) Please change "i.e., haplogroups:" to "haplogroups" (line 431).

265) Please replace "i.e. haplogroups:" with "haplogroups" (line 432).

266) Please change "pathologically is" to "is pathologically" (line 438).

267) Please replace "neuromelanin containing" with "neuromelanin-containing" (line 438).

268) Please change "rigidity, and" to "rigidity and" (line 441).

269) Please replace "[117] ." with "[117]." (line 442).

270) Please change "neurons, but" to "neurons but" (line 448).

271) Please replace "ALZHEIMER DISEASE" with "ALZHEIMER'S DISEASE" in Figure 3.

272) Please realign "SCLEROSIS" horizontally so that it becomes centered under the "AMYOTROPHIC LATERAL" caption in Figure 3.

273) Please change "Alzheimer disease (AD)" to "Alzheimer's disease (AD)." (line 452).

274) Please replace "circle" with "cycle" (line 455).

275) Please change "(PD)" to "(PD)." (line 456).

276) Please change "it has been identified" to "display" (line 457).

277) Please replace "deletions on the mtDNA" with "mtDNA deletions" (line 458).

278) Please change "now" to "now," (line 459).

279) Please replace "are" with "have been" (line 459).

280) Please change "behind" to "underpinning" (line 460).

281) Please define abbreviation for "mtBER" (line 461) and "SLA" (line 535).

282) Please replace "(4)" with "(4)." (line 461).

283) Please change "(ALS)" to "(ALS)." (line 461).

284) Please replace "(1)" with "(1)," (lines 462, 468).

285) "it is documented in ALS the alteration of the nuclear DNA repair system which can support the impairment of the mtBER" is not clear in "As of yet there is no evidence for this hypothesis, even though it is documented in ALS the alteration of the nuclear DNA repair system which can support the impairment of the mtBER (4)" (line 465) for the following reasons:

a) the grammatically correct order seems to be reversed: "the alteration of the nuclear DNA repair system, which can support the impairment of the mtBER, is documented in ALS"

b) it is not exactly clear what the authors mean by "alteration of the nuclear DNA repair system" and "the impairment of the mtBER"

c) it is further not clear how can "alteration of the nuclear DNA repair system" support "the impairment of the mtBER" and why "the nuclear DNA repair system" needs to be altered before it can support "the impairment of the mtBER"?

d) it is also not clear how can "alteration of the nuclear DNA repair system" concretely support "the impairment of the mtBER", for example does this involve its mitochondrial translocation? If so, verbs such as "substitute" or "supplant" might be used in place of "support" to better underscore this notion.

286) Please change "(HD)" to "(HD)." (line 467).

287) Please replace "mutant htt" with "htt" (line 467).

288) Please change "potential and more generally" to "potential, and more generally," (line 469).

289) Please replace "Ca2+ unbalanced" to "imbalanced Ca2+" (line 472).

290) "described" appears twice in "Enlighted complexes represent components of the OXPHOS in which has been described a mutation related to the disease described" (line 473). Please fix.

291) It is not clear what the authors mean by "Enlighted complexes" and "disease described" in "Enlighted complexes represent components of the OXPHOS in which has been described a mutation related to the disease described" (line 473)?

292) "Enlighted complexes represent components of the OXPHOS in which has been described a mutation related to the disease described" might not be grammatically correct with respect to "in which has been described a mutation related to the disease described" (line 473).

293) Please change "Amyloid" to "amyloid" (line 474).

294) Please replace "Base Excision Repair (pathway)" to "base excision repair" (line 474).

295) Please change "Reactive Oxygen Species ." to "reactive oxygen species." (line 475).

296) Please replace "Studies in a rat PD model, employing rotenone treatment that inhibits the complex I of the respiratory chain, demonstrate" with something like "Studies deploying the complex I inhibitor rotenone in a rat PD model, demonstrated" or "Studies deploying the complex I inhibitor rotenone, demonstrated in a rat PD model" (line 476).

297) "Indeed, when complex I is inhibited, midbrain neurons produce more H2O2 compared to cortical neurons and consequently their mtDNA presents more oxidative damage" (line 479) does not seem to be grammatically correct with respect to "mtDNA presents more oxidative damage". Please revise.

298) Please replace "base excision repair" with "BER" (lines 482, 512).

299) Please change "PD" to "PD," (line 489).

300) Please replace "Lateral Sclerosis" with "lateral sclerosis" (line 494).

301) Please change "75-80%" to "75–80%" (line 496).

302) Please replace "symptoms, such" with "symptoms such" (line 497).

303) Please change "gene, codifying" to "gene codifying" (line 499).

304) Please replace "protein" with "enzyme" (line 500).

305) Please change "ALS pathophysiology" to "pathophysiology of ALS" (line 502).

306) Please replace "At" with "In" (line 511).

307) Please change "neurons" to "neuron" (line 513).

308) Please replace "HD, such" with "HD such" (line 518).

309) Please change "between that" to "between" (line 526).

310) Please replace "depletion" with "depletions" (line 528).

311) It is not exactly clear what the authors mean by "L3 higher Aβ42 levels" (L1 and L3), "Not congruent/disparity" (UK, T), and "Not validated relevance and not congruent studies" (UK, K, J, and JT) in Table 2?

312) Please change "neutralise" to "neutralises" (K and U), "Act" to "Acts" (HV, H, HV5), "AD (HV haplogroup also)" to "AD" (H), "Parkinson's disease" to "PD (UKJT and R), "Decreased risk" to "Decreased PD risk" (B) in Table 2.

313) Please replace "genes, directly" with "genes directly" (line 536).

314) Please change "NDs, such as AD, PD, and" to "NDs such as AD, PD and" (line 537).

315) Please replace "play" with "plays" (line 540).

316) Please change "are confirming" to "confirm" (line 548).

317) Please replace "V.B" with "V.B." (lines 552, 553).

318) Please change "L.P" to "L.P." (line 553).

319) Please replace "for" with "with" (line 556).

Round 2

Reviewer 2 Report

Major point:

"Indeed, at early timepoints after DNA damage, cells attempt to preserve as many mitochondria as possible to energy demand; while at later time points, mitophagy increases and cells clear the superfluous or damaged mitochondria produced during OXPHOS" (page 7) seems to be adapted from reference [60] (Dan, X.; Babbar, M.; Moore, A.; Wechter, N.; Tian, J.; Mohanty, J., G, ; Croteau, D., L, ; Bohr, V., A., DNA damage invokes mitophagy through a pathway involving Spata18. Nucleic Acids Res 2020, 48, 6611-6623) "It seems that at early timepoints after DNA damage, the cells attempt to preserve as many mitochondria as possible as a means to support energy demand; while at later time points, mitophagy increases and cells clear the superfluous or damaged mitochondria produced during oxidative phosphorylation process" (page 11). Please rephrase using exclusively your own words.

Minor points:

1) Please change "subsequent signalling cascade" to "subsequent signalling cascades" (page 1).

2) Please replace "homeostasis are called" to something like "homeostasis fall under the umbrella of" (page 2).

3) Please change "(UPS), and the mitochondria-specific unfolded protein response (UPRmt), mitochondrial fusion and fission dynamics, and" to "(UPS), the mitochondria-specific unfolded protein response (UPRmt), mitochondrial fusion and fission dynamics and" (page 2).

4) Please replace "impairs the cellular function and" with "impairs cellular function and" (page 2).

5) Please change "and also exerts a multi-organ" to "and exerts a multi-organ" (page 2).

6) Please replace "by an unique architecture, including" with "by a unique architecture including" (page 2).

7) "that cannot dilute out defective" could be changed to "that cannot efficiently dilute out defective" (page 2).

8) Please change "In agreement" to "In agreement with this" (page 2).

9) Please replace "it promotes the brain decline" to "it promotes brain decline" (page 2).

10) Please change "relevance of altered metabolic function as" to "the relevance of altered metabolic function as a" (page 2).

11) Please replace "In this contest" with "In this context" (page 2).

12) Please change "occurring in ageing and in neurodegeneration" to "occurring during ageing and in neurodegeneration" (page 2).

13) Please replace "In this review we focus" with "In this review, we focus" (page 2).

14) Please change "mitochondria are the only organelle, together with the nucleus, possessing" to "mitochondria, together with the nucleus, are the only organelle possessing" (page 2).

15) Please change "and in turn participate" to "and, in turn, participate" or "and thus participate" (page 2).

16) Please replace "that in turn affect" with "that, in turn, affects" (page 2).

17) It is not exactly clear what the authors mean by "mtDNA homeostasis failure" in "Herein, we describe the effect of mtDNA homeostasis failure in promoting ageing and neurodegenerative processes" (page 2)?

18) Please change "(nDNA), because of mtDNA enhanced" to "(nDNA) because of enhanced mtDNA" (page 2).

19) Please replace "exposure to radical oxygen species (ROS) as O2- or H2O2 which can be produced during the OXPHOS system" with "exposure to radical oxygen species (ROS) such as O2- or H2O2 produced during OXPHOS" (page 2).

20) Please change "system for mtDNA damages" to "system for mtDNA damage" (page 2).

21) "imbalance between production of ROS"/"imbalance in ROS production"/"imbalance between ROS production"/ is mentioned three times in "Oxidative stress represents an imbalance between production of ROS and their elimination by protective mechanisms. In physiological conditions, ROS can activate signalling cascades [15] and are involved in cellular process such as proliferation [16], apoptosis [17] and senescence [18]. Antioxidant enzymes and ROS scavenger systems such as superoxide dismutases (SODs), the thioredoxin system and glutathione peroxidase counteract ROS production, thus preventing the cells from the dangerous effects associated with an imbalance in ROS production [19]. When there is an imbalance between ROS production and the antioxidant systems, oxidative stress occurs and can damage all the main cellular components. Indeed, oxidative stress activates various pathways, including the pro-inflammatory cascade and promotes the formation of pro-mutagenic DNA adducts, creating genetic instability that leads to DNA mutations, which alter cellular homeostasis" (page 2).

22) Please replace "can activate signalling cascades" with "can activate diverse signalling cascades" (page 2).

23) Please change "activates various pathways, including" to "activates various pathways including" (page 2).

24) Please fill in the rest of the row with text following "adducts, creating genetic instability that leads to DNA" in "Indeed, oxidative stress activates various pathways, including the
pro-inflammatory cascade and promotes the formation of pro-mutagenic DNA adducts, creating genetic instability that leads to DNA mutations, which alter cellular homeostasis" (page 2).

25) Please redraw the blue arrows pertaining to "Mitochondrial Dynamics" using exactly the same color hue as in "UPRmt" in Figure 1.

26) Please replace "antioxidants, superoxide dismutase" with "antioxidants, SOD" (page 4)

27) The structure of "If all these responses to the damage are not able to fully repair the lesion, then a molecule of mtDNA can be degraded without impacting mitochondrial physiology, owing to the presence of multiple copies of the same nucleic acid within mitochondria" is too complex (page 4). Please split into two sentences.

28) Please replace "these responses to the damage" to "these damage responses" (page 4).

29) It is not exactly clear what the authors mean by "damage is extended" in "Finally, if the damage is extended, the whole mitochondria can be degraded through mitophagy" (page 4)?

30) Please change "whole mitochondria can be" to "whole mitochondria are" (page 4).

31) It is not exactly clear to what physiological process the authors are referring to as "overflow the proteosome" in "Indeed, dysfunctional mitochondria with an increased amount of damaged proteins could not only overflow the proteosome but also affect the proteasomal subunits themselves, thereby affecting the catalytic activity of the UPS" (page 4)? Please rephrase in scientific terms.

32) Please replace "overflow the proteosome" with "overflow the proteasome" (page 4).

33) From "Once dysfunction and proteasomal impairments develops then a vicious cycle may start, leading to a progressive failure of the system and consequently to ageing or, in the worst scenario, to neurodegenerative diseases" (page 4) is not clear whether the authors are referring to mitochondrial "dysfunction"?

34) It is also not clear what the authors mean by "system" in "Once dysfunction and proteasomal impairments develops then a vicious cycle may start, leading to a progressive failure of the system and consequently to ageing or, in the worst scenario, to neurodegenerative diseases" (page 4)?

35) Please change "inter-/intra-strand" to "intrastrand and interstrand DNA" (pages 4, 7).

36) Please change "a dysfunction in mitochondria" to "mitochondrial dysfunction" (page 4).

37) Please replace "dynamin-relate protein I" with "dynamin-related protein 1" (page 4).

38) Please change "which will be later cleared" to "for clearance" (page 4).

39) Please replace "lysosome give rise to" with "lysosome gives rise to" (page 4).

40) Please change "proteins, lipids, and nucleic" to "proteins, lipids and nucleic" (page 4).

41) Please replace "DSB: double strand break" with "DSB = double strand break" (page 4).

42) Please change "with the 8-oxo-7,8-dihydro-2’-deoxyguanosine" to "with 8-oxo-7,8-dihydro-2’-deoxyguanosine" (page 4).

43) Please replace "by a H-strand" with "by an H-strand" (page 4).

44) Please change "proteins involved in mitochondrial physiology" to "mitochondrial proteins" (page 4).

45) Please replace "the nDNA, but also in the mtDNA" with "nDNA but also in mtDNA" (page 4).

46) Please change "replication by the DNA polymerase" to "replication by DNA polymerase" (page 4).

47) Please replace "rNMPs repair systems" with "rNMP repair systems" (page 4).

48) Please change "alterations of several mechanisms" to "alterations in several mechanisms" (page 5).

49) Although the authors have indicated that it is "DNA repair impairment" inducing mtDNA alternations, whether impairment of "DNA replication" leads to the same result is not explicitly specified in "Deletions in mtDNA derive from alterations of several mechanisms: (i) DNA replication; (ii) DNA repair impairment, resulting in double-strand breaks." (page 5). Please fix.

50) It is not clear what the authors mean by "major arc" and "minor arc" in "The majority of the mtDNA deletions occur in the major arc of the mtDNA and have been associated with different pathologies, where the clinical prognosis directly correlates with the mtDNA deletion frequency. Nonetheless, specific deletions are found also on the minor arc" (page 5)? Please define in the text.

51) Please replace "Single-strand and double-strand" with "Single-strand and double-strand DNA" (page 5).

52) Please change "that can occur on mtDNA" to "that occur on mtDNA" (page 5).

53) Please replace "The presence of these lesions" with "Abundance of these lesions" (page 5).

54) Please change "Base excision repair pathway" to "The base excision repair pathway" (page 5).

55) Please replace "mitochondria is the BER pathway" with "mitochondria was the BER pathway" (page 5).

56) Please change "Oxidative damages caused by ROS are" to "Oxidative damage caused by ROS is" (page 5).

57) Please format "N" in "N-glycosidic" using italics.

58) Please change "(APE1) that hydrolyse the phosphate backbone" to "(APE1), which hydrolyses the phosphate backbone" (page 5).

59) Please replace "It has been described that there" with "There" (page 5).

60) Please change "different DNA alterations: alkylation" to "different DNA alterations, alkylation" (page 5).

61) Please replace "bifunctional glycosylases activity" with "bifunctional glycosylase activity" (page 5).

62) Please change "glycosylases, while oxidised bases" to "glycosylases while oxidised bases" (page 5).

63) It is not exactly clear what the authors are referring to as "sub-pathways" in "When BER is activated, it can follow two sub-pathways: the short-patch or the long-patch BER (SP- or LP-BER)" (page 5)?

64) Please replace "sub-pathways: the short-patch or the long-patch" with "sub-pathways, the short-patch or the long-patch" (page 5).

65) "The main difference is related to the number of nucleotides that are substituted during the correction process and the proteins involved: in the SP-BER only the damaged nucleotide is removed and corrected, while in the LP-BER from 2 up to 8 nucleotides surrounding the damaged base can be substituted during the repair process" (page 5) is too long. Please split into two sentences.

66) Please change "involved: in the SP-BER" to "involved, in the SP-BER," (page 5).

67) Please replace "and corrected, while in the LP-BER" with "and corrected while in the LP-BER," (page 5).

68) Please change "enzymes involved in the process" to "enzymes involved" (page 6).

69) The structure of "Particularly, in mitochondria, only one DNA ligase has been described, DNA ligase III, which is involved in both replication and repair, while DNA ligase I is also present in nuclei" (page 6) is too complex. Please split into two sentences.

70) Please replace "both replication and repair" with "both DNA replication and repair" (page 6).

71) It is not exactly clear what the authors are referring to as "sub-pathway" in "More interesting is the study of LP-BER sub-pathway in mitochondria" (page 6)?

72) The structure of "Until recently, it was believed that only the SP-BER is active within mitochondria, but several studies carried out in the last decade clearly indicate the existence of a mitochondrial LP-BER, where the protein FEN-1 plays a crucial role" (page 6) is too complex. Please split into two sentences.

73) Please change "clearly indicate the existence" to "clearly indicated the existence" (page 6).

74) Please replace "where the protein FEN-1" with "wherein the protein FEN-1" (page 6).

75) Please change "Mismatch repair pathway and double-strand break repair pathways" to "The mismatch repair pathway and double-strand DNA break repair pathways" (page 6).

76) Please replace "the presence of a MMR involving the" with "that MMR involves a" (page 6).

77) Please change "[39] (Table 1)" to "(Table 1) [39]" (page 6).

78) Please replace "the presence of proteins like" with "the presence of proteins such as" (page 6).

79) Please change "confirm the capability" to "confirm the capacity" (page 6).

80) Please replace "suggest that the recombination" with "suggest that recombination" (page 6).

81) Please change "a not-well characterised" to "a not well-characterised" (page 6).

82) "support"/"supported" appears twice in "Interestingly, other reports support a not-well characterised microhomology-mediated end-joining (MMEJ) repair pathway [45-47] rather than the NHEJ, which appears to be undetectable in mitochondria. This hypothesis is supported by the detection of short repetitive sequences flanking the deletions in mtDNA occurring in 85% of older than 55 days Drosophila and in two-thirds of the reported mitochondrial deletions of aging humans [48, 49] and suggests the existence of a recombination mechanism involved in the maintenance of mtDNA integrity" (page 6). Please fix.

83) Please replace "a not-well characterised microhomology-mediated end-joining (MMEJ) repair pathway [45-47]" with something like "the role of a not-well characterised microhomology-mediated end-joining (MMEJ) repair pathway in mtDNA repair [45-47]," (page 6).

84) "This hypothesis is supported by the detection of short repetitive sequences flanking the deletions in mtDNA occurring in 85% of older than 55 days Drosophila and in two-thirds of the reported mitochondrial deletions of aging humans [48, 49] and suggests the existence of a recombination mechanism involved in the maintenance of mtDNA integrity" (page 6) is too long. Please split into at least two sentences.

85) Please change "older than 55 days Drosophila" to "Drosophila older than 55 days" (page 6).

86) "A heteroplasmic cell undergoing mtDNA replication – which is independent from the cell cycle – can lead to an accumulation of mutations over time" (page 6) does not seem to be semantically correct as it is hard to imagine that a "cell" can "lead to an accumulation of mutations". Please revise.

87) Please replace "undergoes the cell cycle" with "undergoes cell cycle" (page 6).

88) Could the authors please find a more descriptive word to replace "interesting" in "The selective depletion of mtDNA is an interesting phenomenon driving the control of the amount of mutated mtDNA inside a cell" (page 6)?

89) Please change "mtDNA inside a cell" to "mtDNA" (page 6).

90) Please replace "there may be thousands of mtDNA molecules" with "there are thousands of copies of mtDNA" (page 6).

91) Please change "some of them does not compromise" to "only some of these molecules does not necessarily compromise" (page 6).

92) Please replace "lesions as DSBs" with "lesions such as DSBs" (page 6).

93) Please change "underlying the mtDNA degradation" to "underlying mtDNA degradation" (page 6).

94) Please replace "mechanisms have been proposed" with something like "scenarios have been proposed" (page 6).

95) "Two mechanisms have been proposed: (i) degradation by nucleases [53] and (ii) elimination of the whole mitochondria carrying the lesion via autophagy or mitophagy" (page 6) is puzzling as it is not clear what is the difference between the degradation of mitochondria via autophagy and mitophagy? Is this not an single pathway?

96) Please change "It seems that mtDNA" to "mtDNA" (page 6).

97) It is not clear what the authors mean by "This mechanism" in "This mechanism is not damage-specific and its kinetics vary depending on the cell type" (page 6)? Are the authors referring to "degradation by nucleases" or "elimination of the whole mitochondria carrying the lesion via autophagy or mitophagy"? Please rephrase unequivocally.

98) Please replace "that linear DNA formed" with "that the linear DNA formed" (page 6).

99) Please change "pol" to "POL" (page 7).

100) Please replace "on mitochondrial integrity" with "of mitochondrial integrity" (page 7).

101) Please change "promoting the DNA damage" to either "promoting the damage", "inducing the damage", or "triggering the damage" (page 7).

102) Please replace "They also underlined" with "They also emphasized" (page 7).

103) Please change "highlighting the mechanism" to "shedding light on the mechanism" (page 7).

104) Please replace "following irreparable mtDNA damages" with "following irreparable mtDNA damage" (page 7).

105) Please explain the concept of "selective mitochondrial fusion" to complement mtDNA damage in detail in the "3.4 Mitochondrial dynamics" section.

106) Please change "fuse and divide in a phenomenon known as fission and fusion" to either "fuse and divide in a phenomenon known as fusion and fission, respectively" or "divide and fuse in a phenomenon known as fission and fusion, respectively" (page 7).

107) "Mitochondria form a dynamic network of organelles able to fuse and divide in a phenomenon known as fission and fusion" (page 7) is too trivial since it is rather self-evident that the ability of organelles to "fuse" is called "fusion".

108) Please replace "by the GTPase dynamin-relate" with "by the GTPase dynamin-related" (page 7).

109) Please change "there are no clear evidence of" to "there is no clear evidence of" (page 7).

110) "Moreover, whether mtDNA damage, fission dynamics are correlative or consequential is also elusive" (page 7) is puzzling since it mentions "fission dynamics" whereas the preceding sentence "Indeed, mtDNA lesions such as pyrymidine dimers, base modifications or inter-/intra-strand and DNA-protein cross-links are cleared by isolating dysfunctional mitochondria and their removal by selective mitochondrial fusion and mitophagy" refers to "mitochondrial fusion". Please reconcile this dichotomy.

111) Please replace "fission dynamics are correlative" with "mitochondrial fission dynamics are correlative" (page 7).

112) Please change "best characterised repair pathway" to "best characterised DNA repair pathway" (page 8).

113) Please replace "the difference to its" with "the difference with its" (page 8).

114) Please change "we report a list of proteins" to "we report proteins" or "we enlist proteins" (page 8).

115) Please move the abbreviation list of Table 1 so that it becomes confluent with the rest of the legend text.

116) Please replace "SSBs = single-strand breaks" with "SSBs = single-strand DNA breaks" (page 8).

117) Please change "5'-3' exonuclease activity" to "5'–3' exonuclease activity" in Table 1 (MGME1).

118) Please replace "antagonistic factors (mitochondrial dysfunctions)" with "antagonistic factors (mitochondrial dysfunction)" (page 9).

119) Please format the first paragraph of the "4.1 Mitochondrial free radical theory of ageing" section in justify (page 10).

120) Please change "Dr Denham Harman" to "Dr. Denham Harman" (page 10).

121) Please replace "mechanisms, which would lead" with "mechanisms leading" (page 10).

122) "In following publications, Harman suggested that these free radicals were produced within mitochondria due to their high oxygen usage and the correlation between basal metabolic rate and ageing, and that ageing originates within mitochondria too, with mtDNA receiving around 16 times more oxidative damage than nDNA" (page 10) is too long. Please split into two sentences.

123) Please change "within mitochondria too" to "within mitochondria as well" (page 10).

124) Please replace "in vivo [84], suggesting" with "in vivo [84] suggesting" (page 10).

125) Please replace "an independent manner to" with "a manner independent of" (page 10).

126) Please change "process well characterised" to "well characterised process" (page 10).

127) Please replace "selected for and yielding" with "selected for yielding" (page 10).

128) Please change "early during the embryonic development" to "early during embryonic development" (page 10).

129) Please replace "This decreased apoptosis" with "Decreased apoptosis" (page 10).

130) "and in fact mitochondrial OXPHOS deficiency is accepted as a hallmark of ageing" does not seem to fit "This decreased apoptosis results in dysfunctional and deficient mitochondria remaining prevalent within cells, promoting premature ageing, and in fact mitochondrial OXPHOS deficiency is accepted as a hallmark of ageing" (page 10). Please rephrase or split into two separate sentences.

131) Please change "As well as promoting ageing" to "In addition to promoting ageing" (page 10).

132) Please replace "homeostasis, caused" with "homeostasis caused" (page 11).

133) From "However, over a lifetime, the body develops ROS-independent damage, which stimulates the cell stress response, thereby elevating the ROS production in a vicious cycle" (page 11) is not explicitly clear whose organism body are the authors referring to?

134) Please change "damage that accelerate" to "damage that accelerates" (page 11).

135) Please replace "ageing-related mechanisms which are participating in the progression of several diseases" with "mechanisms that participate in the progression of age-related diseases" (page 11).

136) "Below, we focus on the role of mtDNA alterations in neurodegenerative diseases" seems to be redundant (page 11).

137) Please change "such as but not exclusively" to "such as, but not exclusively" (page 11).

138) Please replace "problems, due to initial neuronal dysfunction, followed" with "problems due to initial neuronal dysfunction followed" (page 11).

139) Please change "everyday life and as a consequence" to "everyday life as a consequence" (page 11).

140) Please replace "amyotrophic lateral sclerosis" with "amyotrophic lateral sclerosis (ALS)" (page 11) and "Amyotrophic lateral sclerosis (ALS)" with "ALS" (page 14).

141) Please change "several data" to something like "several lines of evidence" (page 12).

142) Please replace "damage in NDs progression" with "damage in ND progression" (page 12).

143) "and AD and PD haplogroups are summarised in Table 2" does not seem to fit "Herein we summarise the up-to-date knowledge about the role of mitochondria and mtDNA damage in NDs progression (Figure 3) and AD and PD haplogroups are summarised in Table 2" (page 12). Please fix.

144) Please change "are familiar, due to" to "are familiar due to" (page 12).

145) Please replace "environment, and epigenetic" with "environment and epigenetic" (page 12).

146) Please change "there are evidences confirming" to "there is evidence confirming" (page 12).

147) Please change "up to now there is" to "up to now, there is" (page 12).

148) Please replace "function of the repair pathway" with "function of the DNA repair pathway" (page 12).

149) The paragraph "Finally, an interesting association has been revealed between mtDNA polymorphisms and AD. Small differences in the encoded proteins can slightly affect the OXPHOS activity, leading to either overproduction or reduction of free radicals. Thus, a polymorphism can predispose individuals to an accumulation of somatic mtDNA mutations, OXPHOS impairment and therefore to an increased or decreased risk of developing AD. Different haplogroups have been identified and related to both an increased or decreased risk of AD [114]. However, these studies are controversial and deeper investigations are required to fully understand the possible role of mtDNA polymorphism in the pathogenesis of AD" does not seem to contain any specific information. Please give at least one concrete example of how mtDNA polymorphism modulates AD pathophysiology.

150) Please change "OXPHOS activity, leading to" to "OXPHOS activity leading to" (page 12).

151) From "Thus, a polymorphism can predispose individuals to an accumulation of somatic mtDNA mutations, OXPHOS impairment and therefore to an increased or decreased risk of developing AD" (page 12) is not explicitly clear whether polymorphism leads to increased or decreased risk of developing AD in individuals?

152) Please replace "rigidity, and bradykinesia" with "rigidity and bradykinesia" (page 12).

153) Please change "mitochondrial dysfunction and mtDNA damages" to "mitochondrial dysfunction and mtDNA damage" (page 12).

154) Please replace "sites on mtDNA of nigral neurons" with "sites in mtDNA of nigral neurons" (page 12).

155) Please change "SOD 1 mutation" to "SOD1 mutation" in Figure 3 (ALS).

156) Please define "SOD1" in the abbreviation list of the legend to Figure 3.

157) Please change "mtDNA damage in neurodegenerative diseases" to "mtDNA damage in neurodegenerative diseases." (page 14).

158) Please replace "complexes I, III, IV, and V" with "complexes I, III, IV and V" (page 14).

159) Please change "promotion of Aβ accumulation (3), and increased" to "promotion of Aβ accumulation (3) and increased" (page 14).

160) Please replace "which in turn exacerbates" with "which, in turn, exacerbates" (page 14).

161) Please change "mtDNA deletions (3), with" to "mtDNA deletions (3) with" (page 14).

162) Please replace "mitochondria of the patients’ neurons" with "mitochondria in patient neurons" (page 14).

163) Please change "affects mitochondrial DNA stability" to "affects mtDNA stability" (page 14).

164) Please replace "mtDNA repair, rather than" with "mtDNA repair rather than" (page 14).

165) Please change "As of yet there" to "As of yet, there" (page 14).

166) Please replace "hypothesis, even though the" with "hypothesis even though the" (page 14).

167) It is not clear what the authors mean by "the alteration of the nDNA repair system supports the impairment of the mitochondrial BER" in "As of yet there is no evidence for this hypothesis, even though the alteration of the nDNA repair system supports the impairment of the mitochondrial BER as documented in ALS" (page 14)? What exactly are the authors referring to by "the alteration of the nDNA repair system" and how does this alteration "support the impairment of the mitochondrial BER"?

168) Please change "[125-127](4)" to "[125-127] (4)" (page 14).

169) Please replace "PGC-1 (1) which negatively" with "PGC-1 (1), which negatively" (page 14).

170) Please change "impacts the ROS scavenging mechanisms" to "impacts ROS scavenging mechanisms" (page 14).

171) Please replace "more generally the whole" with "more generally, the whole" (page 14).

172) From "Correspondingly, ROS production is exacerbated (3), leading to increased mtDNA mutation and depletion, consequently disrupting mitochondrial integrity (4)" (page 14) is not evident under which conditions is ROS production exacerbated? Please fix.

173) It is not clear what the authors mean by "Enlighted complexes" and "disease described" in "Enlighted complexes represent components of the OXPHOS in which a mutation related to the disease described has been reported" (page 14)?

174) Please change "components of the OXPHOS in" to "components of the OXPHOS system, in" (page 14).

175) Please move the abbreviation list of Figure 3 so that it becomes confluent with the rest of the legend text.

176) Please replace "ROS = reactive oxygen species" with "ROS = reactive oxygen species." (page 14).

177) Please change "complex I of the respiratory chain" to "respiratory chain complex I" (page 14).

178) Please replace "PD model, demonstrated that" with "PD model demonstrated that" (page 14).

179) Please change "is inhibited, midbrain neurons produce" to "was inhibited, midbrain neurons produced" (page 14).

180) Please replace "consequently there is increased" with either "consequently leading to" or "consequently resulting in" (page 14).

181) Please change "In agreement" to something like "To this end" (page 14).

182) Please replace "can cause early-onset Parkinsonism and" with "that cause early-onset Parkinsonism" (page 14).

183) Please change "substantia nigra" to "SN" (page 14).

184) Please replace "or degeneration were present" with "or degeneration were reported"  (page 14).

185) "In PD, it has been also observed that there is an accumulation of large deletions in mtDNA and an absence of point mutations [130-132], and it has been hypothesised that a failure in detecting point mutations in late-stage PD tissues might be caused by the degeneration of neurons carrying these mutations at earlier stages of the disease" (page 14) is too long. Please split into two sentences.

186) Please change "In PD, it has been also observed that there is an accumulation of large deletions in mtDNA and an absence of point mutations" to "It has been also observed that there is an accumulation of large deletions in mtDNA and an absence of point mutations in PD patients" (page 14).

187) Please replace "during the adult age" with "during adulthood" (page 14).

188) Please change "lower limbs, while the" to "lower limbs while the" (page 14).

189) Please format "SOD1" in "SOD1 gene codifying" using italics (page 14).

190) Please replace "displays a less protective" with "displays less protective" (page 15).

191) Please change "oxidative stress, resulting" to "oxidative stress resulting" (page 15).

192) Please replace "In agreement" with "Consistently" (page 15).

193) Please change "in ALS, leading to DNA" to "in ALS leading to DNA" (page 15).

194) Please replace "neurons in physiological conditions" with "neurons under physiological conditions" (page 15).

195) Please change "system may be involved" to "system are involved" (page 15).

196) Please format "HTT" in "huntingtin (HTT) gene" using italics (page 15).

197) Please replace "peroxisome proliferation activated" with "peroxisome proliferator-activated" (page 15).

198) Please change "individuals, suggesting a link" to "individuals suggesting a link" (page 15).

199) Please replace "show a reduced activity of mitochondrial complex II and III as well as several mtDNA mutations and depletions" with "show several mtDNA mutations and depletions as well as a reduced activity of mitochondrial complex II and III" (page 15).

200) Please change "Protective effect: neutralises" to "Protective effect, neutralises" (K and U), "No congruent data and disparity between studies" to "Disparity between studies and no congruent data" (UK, T), "Protective: 22% reduction" to "Protective, 22% reduction", and "Protective: reduced risk" to "Protective, decreased PD risk" in Table 2.

201) Please realign "Genetic susceptibility to AD" horizontally so that its formatting matches the rest of the "Activity" cells in Table 2.

202) Please replace "involving the function of mitochondria" with "involving mitochondrial function" (page 16).

203) Please change "AD, PD, and ALS" to "AD, PD and ALS" (page 16).

204) Please replace "metabolism, suggesting that alterations" with "metabolism suggesting that alterations" (page 16).

205) "Alterations in mtDNA are important" does not seem to make sense in "Alterations in mtDNA are important in order to investigate the mechanistic pathways triggered by the environmental and metabolic alterations that are known to be risk factors for sporadic NDs" (page 16). Please either better justify why "Alterations in mtDNA are important" or completely rephrase the sentence.

206) Please change "the results produced up to" to "current results" (page 16).

207) Please replace "that the analysis of such" with "that analyses of such" (page 16).

208) Please change "Naples, Italy, for graphical" to "Naples, Italy for graphical" (page 16).

209) Please replace "no. 956070" with "No. 956070" (page 16).

Round 3

Reviewer 2 Report

Bazzani et al. have coherently summed up the current knowledge on mtDNA repair processes in the context of neurodegeneration. Although definitely not new, this topic is gaining increased traction among the scientific community. Indeed, increased understanding of repair at the mtDNA level can hypothetically push us forward in the direction of custom mtDNA engineering. Although the review paper is overexcessively long and fraught with many instances of "hard-to-read" sections, overall, it leaves a generally good impression on the curious readership. As a bonus, the authors have outsourced the preparation of the cozy figures to a professional illustrator. Hopefully, their manuscript will not only open new avenues of research into mtDNA maintenance and repair in neurodegenerative diseases but also inspire others to prompt investigations into how mtDNA repair processes influence the progression of malignant or metabolic diseases in the wake of aging.

1) Please change "oxidative phosphorylation system (OXPHOS)" to "oxidative phosphorylation (OXPHOS) system" (page 1).

2) Please replace "to the reactive oxygen species" with "to reactive oxygen species" (page 1).

3) Please change "during the ATP production" to "during ATP production" (page 1).

4) Please replace "In this review we summarise" with "In this review, we summarise" (page 1).

5) Please change "the mitochondria repair mechanisms" to "mitochondrial repair mechanisms" (page 1).

6) Please replace "physiological role of mtDNA damages" with "physiological role of mtDNA damage" (page 1).

7) Please change "maintaining the cellular homeostasis" to "maintaining cellular homeostasis" (page 1).

8) Please replace "inter-organelle contact with" with "inter-organelle contacts with" (page 1).

9) Please change "energy demands of the cells" to "energy demands of the cell" (page 1).

10) Please replace "by an unique architecture" with "by a unique architecture" (page 2).

11) Please replace "In this review we focus" with "In this review, we focus" (page 2).

12) Please change "affect the neurons" to "affects the neurons" (page 2).

13) It is not exactly clear what the authors mean by "mtDNA homeostasis failure" in "Herein, we describe the effect of mtDNA homeostasis failure in promoting ageing and neurodegenerative processes" (page 2)?

14) Please replace "2.1. Point mutations and ribonucleotide incorporation" with "2.1 Point mutations and ribonucleotide incorporation" (page 2).

15) Please replace "radical oxygen species (ROS)" with "ROS" (page 2).

16) Please change "H2O2 which can be" to "H2O2, which can be" (page 2).

17) Please replace "during the OXPHOS system" with "during OXPHOS" (page 2).

18) Please replace "system [13], the lack of histones, and" with "system [13] due to the lack of histones and" (page 2).

19) Please change "signalling diverse cascades" to "diverse signalling cascades" (page 2).

20) Please replace "oxidative stress occurs and" with "the resulting oxidative stress" (page 2).

21) Please change "promutagenic DNA adducts, creating" to "promutagenic DNA adducts creating" (page 3).

22) Please replace "prevent or eliminate mtDNA damages" with "prevent or eliminate mtDNA damage" (page 4).

23) Please change "If damage is sensed on the mtDNA, repair mechanisms become activated" to "Repair mechanisms become activated after damage is sensed on the mtDNA" (page 4).

24) Please replace "last decades other common" with "last decades, other common" (page 4).

25) The structure of "If all these responses to the damage are not able to fully repair the lesion, then a molecule of mtDNA can be degraded without impacting mitochondrial physiology, owing to the presence of multiple copies of the same nucleic acid within mitochondria (3)" is rather complicated. Please split into two sentences.

26) Please change "responses to the damage" to "damage responses" (page 4).

27) Please replace "then a molecule of mtDNA can" with "then a single molecule of mtDNA can" (page 4).

28) It is not exactly clear what the authors mean by "damage is extended" in "Finally, if the damage is extended, the whole mitochondria can be degraded through mitophagy" (page 4)?

29) Please change "whole mitochondria can be" to "whole mitochondria are" (page 4).

30) It is not exactly clear to what physiological process the authors are referring to as "overflow the proteosome" in "Indeed, dysfunctional mitochondria with an increased amount of damaged proteins could not only overflow the proteasome but also affect the proteasomal subunits themselves, thereby affecting the catalytic activity of the UPS" (page 4)? Please rephrase in scientific terms.

31) It is not exactly clear what the authors mean by "dysfunction" and "system" in "Once dysfunction and proteasomal impairments develops then a vicious cycle may start, leading to a progressive failure of the system and consequently to ageing or, in the worst scenario, to neurodegenerative diseases" (page 4)?

32) Please change "proteasomal impairments develops then" to "proteasomal impairments develop, then" (page 4).

33) Please replace "crucial role of UPRmt protein" with "crucial role of the UPRmt protein" (page 4).

34) Please change "Under physiological conditions ATF5" to "Under physiological conditions, ATF5" (page 4).

35) Please replace "as characterised in C. elegans" with "such as the one characterised in C. elegans" (page 4).

36) Please change "and it is translocated into the nucleus (blue arrow) where" to "and is translocated into the nucleus (blue arrow), where" (page 4).

37) Please replace "inter-/intra-strand and DNA-proteins cross-links" with "intrastrand and interstrand DNA and DNA-protein cross-links" (pages 4, 7).

38) It is not exactly clear what the authors mean by "damaged part" in "This mechanism can also be used to cope with unrepairable damages such as inter-/intra-strand and DNA-proteins cross-links through the removal of the damaged part by mitophagy" (page 4)?

39) Please change "external stress (3), which" to "external stress (3) that" (page 4).

40) Please replace "causes a mitochondrial dysfunction" with "causes mitochondrial dysfunction" (page 4).

41) Please change "dynamin-relate protein 1" to "dynamin-related protein 1" (page 4).

42) Please replace "works to isolate the damaged" with "acts to isolate the damaged" (page 4).

43) It is not exactly clear what the authors mean by "rest of the cells" in "The whole organelle is isolated from the rest of the cells owing to the generation of an autophagosome" (page 4)?

44) Please change "proteins, lipids, and nucleic acids" to "proteins, lipids and nucleic acids" (page 4).

45) Please change "in a highly controlled manner" to "in a controlled manner" (page 4).

46) Please replace "mitochondria, with 8-oxo-7,8-dihydro-2’-deoxyguanosine" with "mitochondria with 8-oxo-7,8-dihydro-2’-deoxyguanosine" (page 4).

47) Please change "by a H-strand" to "by an H-strand" (page 4).

48) Please replace "spontaneous deamination, giving rise to" with "spontaneous deamination giving rise to" (page 4).

49) Please change "the nDNA but also in the mtDNA" to "nDNA but also in mtDNA" (page 4).

50) Please replace "rNMPs repair systems" with "rNMP repair systems" (page 5).

51) Please replace "alterations in several mechanisms" with "alterations by two mechanisms" (page 5).

52) Although the authors have indicated that it is "DNA repair impairment" inducing mtDNA alternations, whether impairment of "DNA replication" leads to the same result is not explicitly specified in "Deletions in mtDNA derive from alterations in several mechanisms: (i) DNA replication; (ii) DNA repair impairment, resulting in double-strand breaks" (page 5). Please fix.

53) Please change "DNA repair impairment, resulting" to "DNA repair impairment resulting" (page 5).

54) It is not clear what the authors mean by "major arc" and "minor arc" in "The majority of the mtDNA deletions occur in the major arc of the mtDNA and have been associated with different pathologies, where the clinical prognosis directly correlates with the mtDNA deletion frequency. Nonetheless, specific deletions are found also on the minor arc" (page 5)? Please define in the text.

55) Please replace "with the mtDNA deletion frequency" with "with mtDNA deletion frequency" (page 5).

56) Please change "2.3 Single-strand and double-strand breaks" to "2.3 Single-strand and double-strand DNA breaks" (page 5).

57) Please change "3.1. Base excision repair pathway" to "3.1 The base excision repair pathway" (page 5).

58) Please replace "that can occur on mtDNA directly" with "that occur on mtDNA directly" (page 5).

59) Please replace "Oxidative damages caused by ROS are" with "Oxidative damage caused by ROS is" (page 5).

60) Please change "which hydrolyse the phosphate backbone" to "which hydrolyses the phosphate backbone" (page 5).

61) Please replace "monofunctional glycosylases, while oxidised" with "monofunctional glycosylases while oxidised" (page 5).

62) Please change "When BER is activated, it can follow two sub-pathways: the short-patch or the long-patch BER (SP- or LP-BER)" to "When BER is activated, it can follow two sub-pathways, the short-patch or the long-patch BER (SP- or LP-BER)" (page 5).

63) "The main difference is related to the number of nucleotides that are substituted during the correction process and the proteins involved: in the SPBER only the damaged nucleotide is removed and corrected, while in the LPBER from 2 up to 8 nucleotides surrounding the damaged base can be substituted during the repair process." (page 5) is too long. Please split into at least two sentences.

64) Please change "removed and corrected, while in the" to "removed and corrected while in the" (page 5).

65) Please replace "enzymes involved in the process" with "enzymes involved" (page 6).

66) The structure of "Particularly, in mitochondria, only one DNA ligase has been described, DNA ligase III, which is involved in both DNA replication and repair, while DNA ligase I is also present in nuclei" (page 6) is too complex. Please split into two sentences.

67) The structure of "Until recently, it was believed that only the SP-BER is active within mitochondria, but several studies carried out in the last decade clearly indicate the existence of a mitochondrial LP-BER, wherein the protein FEN-1 plays a crucial role" (page 6) is too complex. Please split into two sentences.

68) Please change "last decade clearly indicate" to "last decade clearly indicated" (page 6).

69) Please replace "3.2 Mismatch repair pathway and double-strand break repair pathways" with "3.2 The mismatch repair pathway and double-strand break repair pathways" (page 6).

70) Please change "the presence of a MMR involving the" to "that MMR involves a" (page 6).

71) Please replace "interacts with the glycosylase" with "interacts with the glycosylases" (page 6).

72) It is not exactly clear what the authors mean by "mtDNA is subjected to DSBs" in "mtDNA is subjected to DSBs just as the nDNA, but in mitochondria, the mechanisms involved in DSBs repair are not yet fully elucidated" (page 6)? Please rephrase.

73) Please change "Evidence of mtDSB repair have been found" to "Evidence of mtDSB repair has been found" (page 6).

74) Please define abbreviation for "mtDSB" in "Evidence of mtDSB repair have been found in Drosophila [40] and Saccharomyces cerevisiae" (page 6).

75) It is not exactly clear what the authors mean by "alternate form of Ku80" in "However, the presence of proteins such as XRCC1 [42] or an alternate form of Ku80 [43] in mitochondria is not sufficient to confirm the capacity of mitochondria to repair their mtDSB through homologous recombination (HR) or nonhomologous end joining (NHEJ)" (page 6)?

76) "support"/"supported" is mentioned twice in "Interestingly, other reports support a not-well characterised microhomology-mediated end-joining (MMEJ) repair pathway [45-47] rather than the NHEJ, which appears to be undetectable in mitochondria. This hypothesis is supported by the detection of short repetitive sequences flanking the deletions in mtDNA occurring in 85% of Drosophila older than 55 days and in two-thirds of the reported mitochondrial deletions of aging humans [48, 49] and suggests the existence of a recombination mechanism involved in the maintenance of mtDNA integrity". Please fix.

77) Please replace "a not-well characterised microhomology-mediated end-joining (MMEJ) repair pathway [45-47]" with something like "the role of a not-well characterised microhomology-mediated end-joining (MMEJ) pathway in mtDNA repair [45-47]," (page 6).

78) "This hypothesis is supported by the detection of short repetitive sequences flanking the deletions in mtDNA occurring in 85% of Drosophila older than 55 days and in two-thirds of the reported mitochondrial deletions of aging humans [48, 49] and suggests the existence of a recombination mechanism involved in the maintenance of mtDNA integrity" (page 6) is way too long. Please split into at least two sentences.

79) Please change "can accumulates mutations over time" to "can accumulate mutations over time" or "accumulates mutations over time" (page 6).

80) Please replace "higher than 80%, implying" with "higher than 80% implying" (page 6).

81) "The selective depletion of mtDNA is an interesting phenomenon" sounds rather vague in "The selective depletion of mtDNA is an interesting phenomenon driving the control of the amount of mutated mtDNA" (page 6). Please rephrase.

82) Please change "does not compromise mitochondrial functions" to "does not compromise mitochondrial function" (page 6).

83) "Two scenarios have been proposed: (i) degradation by nucleases [53] and (ii) elimination of the whole mitochondria carrying the lesion via autophagy or mitophagy" is not precise since it suggests the presence of two different mitochondrial degradation pathways, autophagy and mitophagy. Please rephrase so that either it becomes unequivocally clear that mitophagy is a mitochondria-specific autophagy pathway or mention only mitophagy.

84) Please replace "It seems that mtDNA degradation" with "mtDNA degradation" (page 6).

85) It is not clear what the authors mean by "This mechanism" in "This mechanism is not damage-specific and its kinetics vary depending on the cell type" (page 7)? Please ensure that the text can be clearly and unambiguously understood by readers.

86) Please change "DSBs appears to be degraded via" to "DSBs is degraded via" (page 7).

87) Please replace "They also enphasized the" with "They also emphasized the" (page 7).

88) The statement "mitochondria that were produced during OXPHOS" is puzzling in "However, this is in contrast to later timepoints, where the rate of mitophagy is increased, and cells will clear any damaged or unnecessary mitochondria that were produced during OXPHOS" (page 7). How can mitochondria be produced by OXPHOS?

89) "hypothesised" might not well fit the sentence "Overall, it has been hypothesised that following irreparable mtDNA damage, mitochondrial fission is stimulated leading to mitophagy of the damaged mitochondrial daughter" (page 7). Could this word be changed to for example "concluded"?

90) "Mitochondria form a dynamic network of organelles able to fuse and divide in a phenomenon known as fission and fusion, respectively" (page 7) does not make sense for the two following reasons:

a) "fuse and divide in a phenomenon known as fission and fusion, respectively" should read either "divide and fuse in a phenomenon known as fission and fusion, respectively" or "fuse and divide in a phenomenon known as fusion and fission, respectively"

b) that "Mitochondria form a dynamic network of organelles able to fuse" ... "in a phenomenon known as" ... "fusion" is a redundant statement

91) Please change "whose recruitment is controlled by numerous" to "whose mitochondrial recruitment is controlled by" (page 7).

92) Please replace "Fis1, Mff, MiD51 and MiD49" with "Fis1, Mff, MiD49 and MiD51" (page 7).

93) Please change "three GTPase of the dynamin superfamily" to "three GTPases of the dynamin superfamily" (page 7).

94) Please replace "Fission and fusion are critical for" with "Mitochondrial fission and fusion are critical for" (page 7).

95) It is not clear how does "mitochondrial fusion" remove "mtDNA lesions" in "Indeed, mtDNA lesions such as pyrimidine dimers, base modifications or inter-/intra-strand and DNA-protein
cross-links are cleared by isolating dysfunctional mitochondria and their removal by selective mitochondrial fusion and mitophagy [64]. However, the molecular mechanism is still unclear" (page 7)?

96) The statement "removal by selective mitochondrial fusion and mitophagy" is confusing in "Indeed, mtDNA lesions such as pyrimidine dimers, base modifications or inter-/intra-strand and DNA-protein cross-links are cleared by isolating dysfunctional mitochondria and their removal by selective mitochondrial fusion and mitophagy" (page 7) as it might be interpreted such that mitochondrial fusion precedes mitophagy, whereas it is mitochondrial fission that is believed to be imperative for mitochondrial quality control.

97) Please change "mtDNA damage, mitochondrial fission dynamics" to "mtDNA damage and mitochondrial dynamics" (page 7).

98) "Indeed, as well as for fusion, it is not clear if fission starts because of stress and so independently of the mtDNA damage or if it is a consequence of the damage detected on the mtDNA" (page 7) is puzzling for the following reasons:

a) the statement "as well as for fusion" does not seem to refer to any concrete instance in the text of mitochondrial fusion activated by stress

b) the statement "consequence of the damage detected on the mtDNA" relates to "damage detected on the mtDNA", whereas the same concept is true even if mtDNA damage cannot be detected

Please revise both reasonings and simplify the sentence.

99) It is not clear what the authors mean by "delocalisation" in "On the other hand, what is clear is that decreased Drp1 activity and its delocalisation contributes to neurodegeneration by promoting mitochondrial dysfunction" (page 7)?

100) Please replace "what is clear is that decreased Drp1" with "decreased Drp1" (page 7).

101) It is not clear what the authors exactly refer to as "phenomenon" in "More studies are required to elucidate this phenomenon and its involvement in the quality control of mtDNA" (page 7)?

102) Please change "and the difference with its nuclear counterpart" to "and the difference to its nuclear counterpart" (page 8).

103) Please replace "to define the presence of" with something like "elucidate" (page 8).

104) Please change "SSBs = single-strand breaks" to "SSBs = single-strand DNA breaks" (page 8).

105) Please replace "interest in this organelle in the ageing processes" with "interest in the role of this organelle in the ageing process" or "interest in the role of this organelle in ageing" (page 9).

106) It is not clear what the authors mean by "All these approaches" in "All these approaches show both limitations and strengths and a comprehensive
theory explaining the mitochondrial role in ageing is still missing" (page 9)?

107) Please replace "Mitochondrial dysfunctions" with "Mitochondrial dysfunction" in Figure 2 (Integrative factors).

108) "leading to" appears twice in "Harman’s theory starts from the simple observation that free radicals may cause oxidative damage through attacks on cell interior mechanisms leading to degeneration of cells and tissues within the body. This damage could occur to nucleic acids and genetic material, leading to mutations causing cancers, and may also be a contributing factor to ageing" (page 10). Please fix.

109) "In following publications, Harman suggested that these free radicals were produced within mitochondria due to their high oxygen usage and the correlation between basal metabolic rate and ageing, and that ageing originates within mitochondria as well, with mtDNA receiving around 16 times more oxidative damage than nDNA" (page 10) is too long. Please split into two sentences.

110) Please change "non-dividing cells, leading to cellular" to "non-dividing cells leading to cellular" (page 10).

111) Please replace "resistance to stress, resulting in reduced" with "resistance to stress resulting in reduced" (page 10).

112) Please change "selected for and yielding" to "selected for yielding" (page 10).

113) The structure of "This decreased apoptosis results in dysfunctional and deficient mitochondria remaining prevalent within cells, promoting premature ageing, and in fact mitochondrial OXPHOS deficiency is accepted as a hallmark of ageing" (page 10) is too complex. Please split into two sentences.

114) Please replace "and in fact mitochondrial" with "and, in fact, mitochondrial" (page 10).

115) Please change "proliferating cells, while also inducing" to "proliferating cells while also inducing" (page 11).

116) Please replace "4.3 Gradual ROS response theory" with "4.3 The gradual ROS response theory" (page 11).

117) Please either specify "body" as pertaining to a particular organism in "However, over a lifetime, the body develops ROS-independent damage, which stimulates the cell stress response, thereby elevating the ROS production in a vicious cycle" (page 11) or in the preceding text, or rephrase the sentence.

118) Please change "elevating the ROS production" to "elevating ROS production" (page 11).

119) Please replace "peripheral nervous system (PNS)" with "peripheral nervous (PNS) system" (page 11).

120) Please change "which in turn exacerbates" to "which, in turn, exacerbates" (page 12).

121) Please replace "oxidative stress, promoting a vicious circle" with "oxidative stress promoting a vicious circle" (page 12).

122) Please change "DNA ligase III activity, suggesting an" to "DNA ligase III activity suggesting an" (page 12).

123) From "Thus, a polymorphism can predispose individuals to an accumulation of somatic mtDNA mutations, OXPHOS impairment and therefore to an increased or decreased risk of developing AD" (page 12) is not clear whether "a polymorphism can predispose individuals" to increased or decreased "risk of developing AD"?

124) Please replace "rigidity, and bradykinesia" with "rigidity and bradykinesia" (page 12).

125) Please change "CA+2" with "Ca2+" in Figure 3 (Huntington's disease).

126) It is not clear what the authors mean by "the alteration of the nDNA repair system" in "As of yet, there is no evidence for this hypothesis even though the alteration of the nDNA repair system supports the impairment of the mitochondrial BER as documented in ALS (4)" (page 14)?

127) It is not clear what the authors refer to by "Enlighted complexes" and "disease described" in "Enlighted complexes represent components of the OXPHOS system, in which a mutation related to the disease described has been reported" (page 14)?

128) Please replace "SOD1 = superoxide dismutases 1" with "SOD1 = superoxide dismutase 1" in the legend to Figure 3.

129) Please change "in the midbrain neurons, suggesting that" to "in the midbrain neurons suggesting that" (page 14).

130) Please replace "midbrain neurons produce more" with "midbrain neurons produced more" (page 14).

131) Please change "mitochondrial dysfunction or degeneration were reported" to "mitochondrial dysfunction or degeneration were found" (page 14).

132) "It has been also observed that there is an accumulation of large deletions in mtDNA and an absence of point mutations in PD patients [130-132], and it has been hypothesised that a failure in detecting point mutations in late-stage PD tissues might be caused by the degeneration of neurons carrying these mutations at earlier stages of the disease" (page 14) is too long. Please split into at least two sentences.

133) Please replace "neurotoxic, leading to the symptoms" with "neurotoxic leading to the symptoms" (page 15).

134) Please change "such as motor dysfunctions" to "such as motor dysfunction" (page 15).

135) Please replace "peroxisome proliferation-activated" with "peroxisome proliferator-activated" (page 15).

136) Please change "(PGC-1), leading to mitochondrial" to "(PGC-1) leading to mitochondrial" (page 15).

137) Please replace "transgenic HD mice, suggesting that" with "transgenic HD mice suggesting that" (page 15).

138) Please format "Haplogroups identified for AD and PD." using bold (page 15).

139) Please change "Protective effect: neutralises" to "Protective effect, neutralises", "UK, K, J, and JT" to "UK, K, J and JT", "Preventive. Resistance against PD" to "Preventive, resistance against PD", "Protective: 22% reduction" to "Protective, 22% reduction", "Protective: decreased" to "Protective, decreased" in Table 2.

140) Please replace "AD, PD, and ALS" with "AD, PD and ALS" (page 16).

141) Please change "that are not carrying any mutation" to "that do not carry any mutation" (page 16).

142) "Alterations in mtDNA are important" does not make sense in "Alterations in mtDNA are important in order to investigate the mechanistic pathways triggered by the environmental and metabolic alterations that are known to be risk factors for sporadic NDs" (page 16). Please fix.
